# LLM DNA: Tracing Model Evolution via Functional Representations

**Zhaomin Wu**[1], **Haodong Zhao**[2], **Ziyang Wang**[1], **Jizhou Guo**[3], **Qian Wang**[1], **Bingsheng He**[1]

[1]Department of Computer Science, National University of Singapore
[2]School of Computer Science, Shanghai Jiao Tong University
[3]Zhiyuan College, Shanghai Jiao Tong University

zhaomin@nus.edu.sg, zhaohaodong@sjtu.edu.cn, wangziyang@u.nus.edu,
sjtu18640985163@sjtu.edu.cn, e0493632@u.nus.edu, dcsheb@nus.edu.sg

## Abstract

The explosive growth of large language models (LLMs) has created a vast but opaque landscape: millions of models exist, yet their evolutionary relationships through fine-tuning, distillation, or adaptation are often undocumented or unclear, complicating LLM management. Existing methods are limited by task specificity, fixed model sets, or strict assumptions about tokenizers or architectures. Inspired by biological DNA, we address these limitations by mathematically defining *LLM DNA* as a low-dimensional, bi-Lipschitz representation of functional behavior. We prove that LLM DNA satisfies *inheritance* and *genetic determinism* properties and establish the existence of DNA. Building on this theory, we derive a general, scalable, training-free pipeline for DNA extraction. In experiments across 305 LLMs, DNA aligns with prior studies on limited subsets and achieves superior or competitive performance on various tasks. Beyond these tasks, DNA comparisons uncover previously undocumented relationships among LLMs. We further construct the evolutionary tree of LLMs using phylogenetic algorithms, which align with shifts from encoder-decoder to decoder-only architectures, reflect temporal progression, and reveal distinct evolutionary speeds across LLM families.

## 1 Introduction

The proliferation of large language models (LLMs) (Zhao et al., 2023) has produced a vast and complex landscape of models. Hugging Face (Hugging Face, 2025) alone hosts millions of models spanning diverse families, architectures, and tokenizers, with the number increasing at an accelerating pace (Rahman et al., 2025). Amid this rapid growth, tracing the evolutionary paths of LLMs through fine-tuning, distillation, or adaptation is critical across several areas: **safety auditing**—tracking how security risks such as backdoors are transferred between LLMs (Cheng et al., 2024); **model governance**—verifying that adaptations and fine-tuning follow license requirements (Duan et al., 2025; Lee et al., 2025; Xu et al., 2025); and **multi-agent systems**—guiding the structure design and planning (Han et al., 2024). This motivates a central research question:

*Can we develop a principled framework to define and analyze the "DNA" of LLMs, enabling us to systematically characterize their functional similarities and evolutionary relationships?*

Existing approaches to defining and extracting LLM "DNA" remain premature. Most prior work compresses LLMs into task-specific representations for model routing (Ong et al., 2025) or ensemble learning (Huang et al., 2024). These representations are trained for specific objectives and neither generalize across tasks nor capture overall model characteristics. Zhuang et al. (2025) propose EmbedLLM, the first task-agnostic approach, which learns a compact representation for multiple downstream tasks. However, this representation is defined relative to a fixed set of LLMs—it changes when new models are added—and thus is not an intrinsic DNA of each model. Nikolic et al. (2025) and Zhu et al. (2025) further explore how to measure provenance or independence between two LLMs by calculating token similarity or parameter correlation, but these methods do not gen-

---

🌐 dna.xtra.science    ⬡ github.com/Xtra-Computing/LLM-DNA    🐍 pypi.org/project/llm-dna

eralize to diverse LLMs with varying architectures and tokenizers. Collectively, these approaches do not furnish a formal and general definition of LLM DNA or a practical method for its extraction.

To address this gap, we introduce the concept of *LLM DNA*: a compact, low-dimensional representation of a model's functional behavior. By analogy to biological DNA, it should satisfy two foundational properties: *inheritance*—small perturbations to the model do not alter the DNA; and *genetic determination*—models with similar DNA exhibit similar characteristics. If such a DNA is well defined and extractable, established phylogenetic tools can be leveraged to analyze LLM evolution.

The challenges of defining and extracting of LLM DNA are two-fold. The theoretical challenge includes mathematically formalizing a DNA that satisfies both properties—inheritance and genetic determination—and proving the existence of DNA under such definition. The practical challenge is to design, given this definition, a general and scalable framework that efficiently extracts DNA across heterogeneous LLMs with distinct architectures, tokenizers, and prediction paradigms.

In this paper, we formally define LLM DNA as a low-dimensional vector that obtained via a bi-Lipschitz map from the LLM functional space. We prove that this definition naturally satisfies *inheritance* and *genetic determination* and, using the Johnson–Lindenstrauss (JL) lemma (Johnson et al., 1984), formally establish the existence of such DNA. To practically extract LLM DNA, we derive random linear projection as an effective extraction method from these theoretical foundations and design *RepTrace*—a general, scalable, training-free pipeline for DNA extraction. Finally, we build the evolutionary tree for 305 LLMs using RepTrace and report various new findings.

**Contributions.** (1) We formally define LLM DNA and prove that it satisfies *inheritance* and *genetic determination* properties, proving the existence of LLM DNA. (2) Building on these results, we propose a general, scalable, training-free pipeline for extracting LLM DNA. (3) Experiments on 305 models show that LLM DNA outperforms existing baselines on specific tasks and aligns with prior findings. (4) Using LLM DNAs, we construct a phylogenetic tree that aligns with known architectural and temporal shifts and reveals undocumented relationships among LLMs.

## 2 RELATED WORK

LLM DNA relates to prior work using terms such as *representation* and *fingerprint*. We group this literature into two views: (i) a *macroscopic* view, which manages large groups of models by mapping each model to a *representation* for model routing and ensemble learning; and (ii) a *microscopic* view, which uses *fingerprints* to assess whether two LLMs are related for provenance or identity.

**Representation of LLMs.** This direction manages large LLM collections by encoding each LLM into a representation. Most methods are task-specific: Ong et al. (2025); Wang et al. (2024) learn representations for model routing, while Huang et al. (2024); Fang et al. (2024) encode models for ensemble learning. For instance, HybridLLM (Ding et al., 2024), trains a binary "easy-hard" classifier to route queries to a small or a larger model. RouteLLM (Ong et al., 2025) learns a supervised router that selects between stronger and weaker models based on query features. MetaLLM (Nguyen et al., 2024) formulates routing as a multi-armed bandit, adaptively trading off expected answer quality against invocation cost via online feedback. LLM-ensemble (Fang et al., 2024) learns the weights for different LLMs to predict a specific attribute value. These task-specific approaches rely on supervised training for particular tasks, making their learned representations not generalizable across tasks. Zhuang et al. (2025) propose EmbedLLM, the first compact embedding supporting multiple tasks (e.g., model routing and benchmark prediction), but it is defined over a fixed model set; adding models requires retraining and can shift the learned embeddings. Yax et al. (2025) propose PhyloLM, a state-of-the-art approach for constructing LLM phylogenies. It defines inter-model distances in a tokenizer-aware manner: for models with the same tokenizer, it uses Nei's genetic distance over output token distributions; for different tokenizers, it approximates distance using the first four characters of concatenated tokens. As a result, the cross-tokenizer metric is dominated by early-token behavior and may not reflect sentence-level distributions. In contrast, our *LLM DNA* is a stable and generalizable sentence-level intrinsic property extractable independently for each LLM.

**Fingerprint of LLMs.** This line of work evaluates dependence or similarity between pairs of LLMs by mapping each model to a representation, or *fingerprint*. Unlike *watermarks*, which actively insert a fingerprint during training (Zhao et al., 2024; Liu et al., 2024; Tang et al., 2025), fingerprinting is post hoc and analyzes identifiable model properties without modifying training. Prior studies

verify whether a remote LLM API serves the claimed model using strategic queries (Pasquini et al., 2025) or specialized prompts (Tsai et al., 2025) with response classification; Fu et al. (2025) further trains classifiers to capture stylistic signatures. Others go beyond text, leveraging parameters or token embeddings for provenance prediction (Nikolic et al., 2025; Zhu et al., 2025). However, these pairwise methods are often structure-dependent and hard to scale to heterogeneous LLMs. LLM DNA, however, is a general representation capturing structural relationships across diverse models.

# 3 DEFINITION AND EXISTENCE OF LLM DNA

This section provides a formal definition of LLM DNA in Section 3.1. Furthermore, Section 3.2 proves the existence of DNA under this definition. Additionally, Appendix A.1 elaborates that this definition satisfies two properties analogous to those of biological DNA.

## 3.1 DEFINITION OF LLM DNA

We formally define an LLM as a function from a discrete sequence of input texts to a vector of continuous logits (Definition 3.1). This definition applies to general LLMs with diverse architectures and adopts an end-to-end perspective. For the theoretical analysis, we assume the input length is finite, i.e., bounded by a constant $m$. Notably, "finiteness" does not imply smallness; the sets involved may still be arbitrarily large. Meanwhile, our practical stochastic-approximation pipeline in Section 4 does not depend on any particular value of $m$. Under this bound, the input space becomes finite, which directly yields Lemma A.4, a key step in establishing the existence of LLM DNA.

**Definition 3.1** (Large Language Model). *Let $\Sigma$ be a finite character set. Let $\Sigma^*$ be the set of all sequences of characters (strings) from $\Sigma$. For a maximum sequence length $m \in \mathbb{Z}^+$, we define the space of admissible sequences as $\mathcal{S}_m := \{x \in \Sigma^* \mid |x| \leq m\}$. Let $N = |\mathcal{S}_m|$. A **Large Language Model (LLM)** is a function $f : \mathcal{S}_m \to \mathcal{O}$ that maps an admissible input sequence to a vector of real-valued logits, with each logit corresponding to a possible output sequence in $\mathcal{S}_m$. We denote the output space of LLM as $\mathcal{O} := \mathbb{R}^N$. The set of all LLMs constitutes the LLM function space, $\mathcal{F}$.*

**Definition 3.2** (DNA of an LLM). *Let $(\mathcal{F}, d_H)$ be the metric space of LLM functions. The **DNA of an LLM** $f \in \mathcal{F}$ is a low-dimensional vector, denoted $\tau_f \in \mathcal{D}$, where the DNA space $\mathcal{D} = \mathbb{R}^L$ is equipped with the standard Euclidean distance $d_\tau(\tau_1, \tau_2) = \|\tau_1 - \tau_2\|_2$. The mapping from $\mathcal{F}$ to $\mathcal{D}$ must satisfy a bi-Lipschitz condition; that is, there exist constants $c_2 > c_1 > 0$ such that for any two LLMs $f_1, f_2 \in \mathcal{F}$ with corresponding DNAs $\tau_{f_1}$ and $\tau_{f_2}$, it holds that: $c_1 \cdot d_H(f_1, f_2) \leq d_\tau(\tau_{f_1}, \tau_{f_2}) \leq c_2 \cdot d_H(f_1, f_2)$*

Definition 3.2 guarantees that functionally similar LLMs have proximate DNAs, and vice versa. These lower and upper bounds in the bi-Lipschitz condition naturally guarantee two key properties of LLM DNA: (1) **Inheritance**—a lineage operation on an LLM, such as fine-tuning, produces descendants with similar DNAs; and (2) **Genetic determinism**—LLMs with similar DNAs exhibit similar functional behaviors. Both properties are formally stated and proved in Appendix A.1.

## 3.2 EXISTENCE OF LLM DNA

This section provides a formal proof for the existence of the LLM DNA. The core idea is that an LLM's function can be represented as a vector in a high-dimensional Hilbert space (Lemma A.4). This allows us to apply the Johnson-Lindenstrauss (JL) Lemma (Johnson et al., 1984), a powerful result in dimensionality reduction, to guarantee the existence of a low-dimensional embedding that preserves the geometry of the LLM functional space $\mathcal{F}$.

**Theorem 3.3** (Existence of LLM DNA). *For any finite set of $K$ LLMs $\mathcal{F}_K = \{f_1, \ldots, f_K\} \subset \mathcal{F}$, a DNA representation (Definition 3.2) with DNA space $\mathcal{D} = \mathbb{R}^L$ exists, satisfying for all $f_i, f_j \in \mathcal{F}_K$, $c_1 \cdot d_H(f_i, f_j) \leq d_\tau(\tau_{f_i}, \tau_{f_j}) \leq c_2 \cdot d_H(f_i, f_j)$, where the target dimension $L$ is $L = O\left(\left[(c_2 + c_1)/(c_2 - c_1)\right]^2 \log K\right)$.*

The proof of Theorem 3.3 appears in Appendix A.3. The required DNA dimension $L$ trades off with the bi-Lipschitz constants $c_1, c_2$, whose tightness—captured by $\epsilon = (c_2 - c_1)/(c_2 + c_1)$—governs embedding distortion. High fidelity (small distortion) requires $c_1 \approx c_2$, yielding very small $\epsilon$ and,

by $L = O(\epsilon^{-2} \log K)$, a large $L$. Conversely, tolerating more distortion (a larger gap between $c_1$ and $c_2$) increases $\epsilon$ and permits a more compact, lower-dimensional DNA representation.

Theorem 3.3 not only guarantees the existence of an LLM DNA but also yields a practical construction. The proof leverages the constructive nature of the JL lemma: the required linear map $E$ need not be specially engineered; a scaled **random linear projection** satisfies the desired bi-Lipschitz property with arbitrarily high probability. This insight is formalized in the following corollary.

**Corollary 3.4** (Construct LLM DNA via Random Projection). *For any finite set of $K$ LLMs $\mathcal{F}_K = \{f_1, \ldots, f_K\} \subset \mathcal{F}$, the LLM DNA in Theorem 3.3 can be obtained from a random linear projection $E : \mathcal{F} \to \mathbb{R}^L$ with probability at least $1 - 2/K^2$.*

Beyond the probability bound established in Corollary 3.4, Larsen & Nelson (2014) proved that random linear projection is the **optimal** linear dimensionality reduction method. Leveraging both its optimality and computational efficiency, we adopt random Gaussian projection for extracting LLM DNA, with practical details introduced in Section 4.

## 4 EXTRACT LLM DNA

Extracting LLM DNA in practice entails two challenges: (i) instantiating the output space $\mathcal{O}$ to reflect **semantic content**, since surface-form string comparisons are brittle; and (ii) evaluating functional distance over the full input space $\mathcal{S}_m$, which is **computationally prohibitive**. We propose an extraction pipeline, *RepTrace*, which addresses these via a semantic response representation (Section 4.1) and an expectation-based functional distance that admits efficient approximation (Section 4.2). The full RepTrace pipeline is provided in Section 4.3 and illustrated in Figure 1.

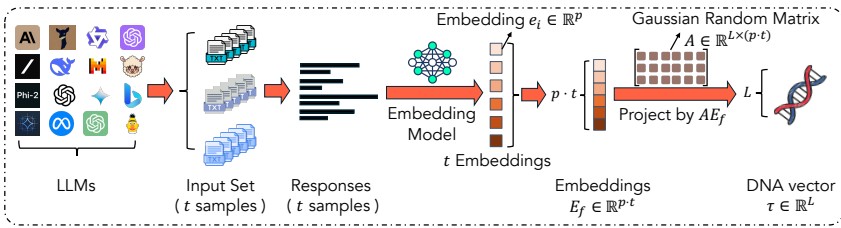

Figure 1: Visualization of RepTrace: LLM DNA extraction workflow

### 4.1 CONSTRUCT SEMANTIC-AWARE REPRESENTATION

This subsection details how to build a semantic-aware vector representation $\phi(f(x_i))$ for each LLM response $x_i$ such that $\|\phi(f_1(x_i)) - \phi(f_2(x_i))\|_2$ effectively approximates $\|f_1(x_j) - f_2(x_j)\|_2$. Purely string-based comparisons ignore semantic information: for example, the words "holiday" and "vacation" are semantically similar yet differ completely as character sequences. To convert textual outputs into vectors that capture semantic meaning, we use a sentence-embedding model to map each response to a fixed-size embedding.

This design has three benefits: (1) it is LLM-agnostic for text-generation models, requiring no access to internal architectures or tokenizers as in many embedding/model-based methods (Yax et al., 2025); (2) it applies to closed-source LLMs because API-only models can still be queried for DNA computation; and (3) it accommodates new LLMs without recomputing existing DNAs, overcoming the limitation of EmbedLLM (Zhuang et al., 2025).

### 4.2 APPROXIMATE LLM FUNCTIONAL DISTANCE

This subsection details our practical approach to measuring the distance between LLMs. Though direct computing functional distance is theoretically sound, its computation requires evaluating functions over the entire, combinatorially large input space $\mathcal{S}_m$, which is infeasible. To bridge theory and practice, we introduce a tractable alternative: a *Stochastic Functional Distance*. This new metric is defined not over the full input space, but as a statistical expectation over a random sample of inputs, reflecting the practical scenario of evaluating models on a finite set of representative prompts.

**Definition 4.1** (Stochastic Functional Distance). *Let $f_1, f_2 \in \mathcal{F}$ be two LLM functions. Let $\mu$ be a probability measure on the input space $\mathcal{S}_m$, and let $t \in \mathbb{Z}^+$ be a fixed sample size. Let $\mathcal{S}_t = \{x_1, \ldots, x_t\}$ denote a set of $t$ independent and identically distributed (i.i.d.) random variables representing inputs drawn from the distribution $\mu$. The **Stochastic Functional Distance** $d_f(f_1, f_2)$ is defined as the expected Euclidean distance between the concatenated semantic representations of the LLMs' outputs over a random sample $\mathcal{S}_t$: $d_f(f_1, f_2) := \mathbb{E}_{\mathcal{S}_t \sim \mu} \left[ \sqrt{\sum_{x_j \in S_t} \|f_1(x_j) - f_2(x_j)\|_2^2} \right]$.*

A concentration bound provides a formal reliability guarantee for our empirical distance. The theorem below shows that the estimate concentrates sharply around the true Hilbert distance $d_H$: for any deviation level, the tail probability decays exponentially in the sample size $t$. Thus, the estimate offers a high-confidence measure of the underlying functional distance.

**Lemma 4.2** (Concentration of Empirical Functional Distance). *Let $f_1, f_2 \in \mathcal{F}$ be two LLM functions. Let $\hat{d}_f^2 = \sum_{j=1}^t \|f_1(x_j) - f_2(x_j)\|_2^2$ be the squared empirical functional distance calculated from a sample of $t$ i.i.d. inputs $\{x_1, \ldots, x_t\}$. Let $d_H^2$ be the true squared Hilbert distance. Assume the squared Euclidean distance between any two output vectors is bounded such that $\|f_1(x) - f_2(x)\|_2^2 \leq C_{\max}$ for any input $x$ and for all functions of interest. Then, for any $\epsilon > 0$, the following bound holds for the sample average: $P \left( \left| \frac{1}{t} \hat{d}_f^2 - d_H^2 \right| \geq \epsilon \right) \leq 2 \exp \left( -\frac{2t\epsilon^2}{C_{\max}^2} \right)$.*

The proof of Lemma 4.2 is included in Appendix A.4. Based on the theoretical bound, we employ the empirical estimate in Definition 4.1 to compute the functional distance in practice. The process begins by sampling a set of $t$ representative prompts, $\mathcal{S}_t = \{x_1, \ldots, x_t\}$, from a distribution $\mu$ that reflects real-world inputs. For a given LLM $f$, we collect the textual response for each prompt $x_j \in \mathcal{S}_t$ and encode it into a semantic vector as described in Section 4.1. These $t$ response vectors are then concatenated end-to-end. This procedure yields a single, high-dimensional vector that serves as a functional representation for the LLM over the sampled inputs. The squared Euclidean distance between two such representations approximates the squared empirical functional distance $\hat{d}_f^2$, which is further a robust and computable estimation of the true squared Hilbert distance $d_H^2$.

### 4.3 Overall Pipeline

This section presents RepTrace algorithm, a training-free end-to-end DNA extraction pipeline. The procedure is detailed in Algorithm 1 and visualized in Figure 1. We begin by selecting a sample input set $\mathcal{S}_t$ of $t$ prompts drawn from real-world datasets. Each input $x_i$ is fed to an LLM $f$ to generate a textual response $y_i$ (line 4), which is then encoded into a fixed-size semantic vector $e_i$ using a sentence-embedding model (line 5). These $t$ embedding vectors are then concatenated to form a single, high-dimensional functional representation $E_f$ for the LLM (line 6). Finally, a pre-computed random Gaussian projection $A$ (line 1) linearly projects this high-dimensional representation $E_f$ into a low-dimensional vector $\tau_f \in \mathbb{R}^L$ (line 7), referred to as the DNA of the LLM.

---

**Algorithm 1:** RepTrace: LLM DNA Extraction Pipeline

---

**Input** : Max length $m$; sample size $t$; DNA dimension $L$; sentence-embedding model $\phi : \Sigma^* \to \mathbb{R}^p$;
A sample input set $\mathcal{S}_t = \{x_1, \ldots, x_t\}$ drawn uniformly from real-world datasets;
A collection of LLMs $\mathcal{F}_K = \{f_1, \ldots, f_K\} \subset \mathcal{F}$.
**Output:** DNA vectors $\{\tau_f \in \mathbb{R}^L : f \in \mathcal{F}_K\}$.

1   **Initialize random Gaussian projection:** draw $A \in \mathbb{R}^{L \times (p \cdot t)}$ with $A_{jk} \sim \mathcal{N}(0, 1/\sqrt{L})$.
2   **foreach** $f \in \mathcal{F}_K$ **do**
3     **for** $i \leftarrow 1$ **to** $t$ **do**
4       $y_i \leftarrow f(x_i)$            `// Textual response of f on input xᵢ`
5       $e_i \leftarrow \phi(y_i) \in \mathbb{R}^p$         `// Semantic embedding`
6     $E_f \leftarrow [e_1, e_2, \ldots, e_t] \in \mathbb{R}^{p \cdot t}$     `// Concatenate response embeddings`
7     $\tau_f \leftarrow A\, E_f \in \mathbb{R}^L$          `// Project into DNA space`
8   **return** $\{\tau_f : f \in \mathcal{F}_K\}$

---

# 5 EXPERIMENTS

This section provides a rigorous empirical validation of LLM DNA as a robust functional fingerprint. Following our experimental setup across 305 models (§5.1, §B), we demonstrate DNA's efficacy in relationship detection (§5.2) and training-free model routing (§5.3). We verify the representation's stability through stress tests involving data heterogeneity (§5.4), synthetic random inputs (§5.5), chat template variations (§5.6), and fine-tuning intensity (§5.7). Systematic ablations characterize the influence of bottleneck embedding models (§C.3), pre-aggregation dimensions $p$ (§C.4), and DNA length $L$ (§C.5). To address technical edge cases, we analyze parameter-size bias (§C.6), same-tokenizer performance (§C.7), and relationship "false positives" (§C.2). We conclude by leveraging these functional distances to reconstruct a phylogenetic tree of LLMs (§5.8, §C.1), providing a data-driven map of the field's architectural evolution.

## 5.1 EXPERIMENTAL SETTING

**Datasets.** We evaluate across a diverse set of datasets covering question answering, commonsense reasoning, and general natural language understanding. Specifically, the pipeline integrates six widely used benchmarks, including SQuAD (Rajpurkar et al., 2016), CommonsenseQA (Talmor et al., 2018), HellaSwag (Zellers et al., 2019), Winogrande (Sakaguchi et al., 2020), ARC-Challenge (Clark et al., 2018), and MMLU (Hendrycks et al., 2021). DNA extraction employs a mixture of all six datasets, with each dataset contributing 100 random samples. Besides, EmbedLLM (Zhuang et al., 2025) dataset is also used to evaluate model routing, though not involved in DNA extraction. For each dataset, prompts are drawn from their designated text fields (e.g., `question`, `sentence`, or `ctx`).

**LLMs.** This study spans 305 LLMs released by 153 organizations (e.g. Qwen, Llama, etc.) on Hugging Face. The evaluation focuses on text-generative models, covering both decoder-only and encoder-decoder architectures, with parameter scales ranging from 100M to 70B. Both instruction-tuned or base models are included in the evaluation. Text embeddings are extracted with the state-of-the-art open-source encoder `Qwen/Qwen3-Embedding-8B` by default, with performance comparison to `sentence-transformers/all-mpnet-base-v2` and `BAAI/bge-large-en-v1.5` presented. More model details are included in Appendix B.

## 5.2 DETECT LLM RELATIONSHIPS

In this subsection, we validate DNA against Zhu et al. (2025), scale to 305 LLMs to demonstrate accurate relationship detection, and visualize the distribution to uncover undocumented model pairs.

Figure 2 shows the DNA distribution of the eight models (four correlated, four independent) in Table 1 of Zhu et al. (2025). The decision boundary is computed by a support vector machine (SVM) with an RBF kernel, indicating that the DNAs of correlated and independent models are clearly separated. This result shows that the DNA structure is consistent with both the experiments in Zhu et al. (2025) and the corresponding Hugging Face documentation.

Table 1 extends this experiment to 305 models. Using the official Hugging Face relationship (the "Model Tree") as ground truth for correlated models, we obtain 83 correlated pairs. We additionally sample 83 random pairs and treat them as independent. The resulting dataset is split 8:2 into training and test sets. Three baselines are compared alongside our approach: *Random*—uniformly guessing independent or correlated; *Greedy*—predicting that all within-organization pairs are correlated and cross-organization pairs are independent; and PhyloLM (Yax et al., 2025)—genetic (Nei) distances to other LLMs in the training set are regarded as a signature vector. For PhyloLM and DNA, an SVM with RBF kernel is used for prediction. From Table 1, three observations emerge. First, DNA significantly outperforms the baselines and achieves a high AUC of 0.992, indicating accurate detection of relationships among heterogeneous LLMs. Second, DNA attains higher recall than precision; this does not necessarily indicate a flaw in DNA, but rather suggests the presence of undocumented relationships in the official model tree. A detailed case study of the misclassified pairs appears in Appendix C.2. Third, DNA quality is robust to the choice of sentence-embedding model. Using BGE (0.3B) and MPNet (0.1B), despite their much smaller parameter sizes, yields comparable—and in this relation prediction task, even better—performance.

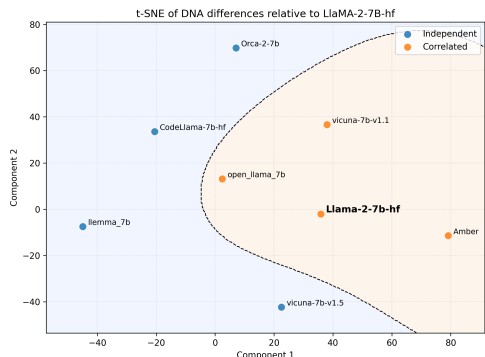

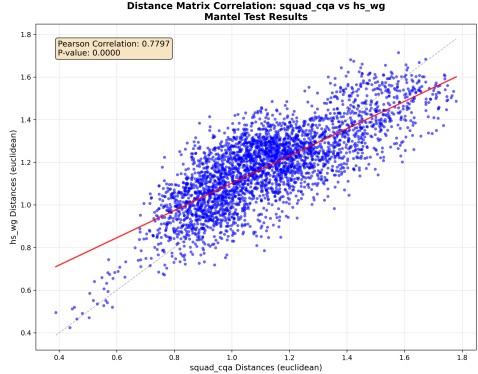

Figure 2: DNA distribution of LLMs evaluated by Zhu et al. (2025). "Independent" and "Correlated" relative to Llama-2-7B-hf are based on public documents. The boundary is computed by an SVM with RBF kernel, indicating that the DNAs of "Independent" and "Correlated" models are clearly separated.

Figure 3: Mantel test between DNA extracted from two disjoint datasets. Each point represents a single pair of models, plotted by their distance in the first dataset versus their distance in the second, showing a strong correlation (Pearson-$r = 0.7797$) and high statistical significance ($p$-value $< 0.0001$).

Figure 4 presents the t-SNE visualization of all 305 models, resulting in three main observations. First, models from the same organization and model family cluster tightly (e.g., Qwen3, Llama), confirming that DNA can be used to detect relationships among LLMs. Second, fine-tuned models are closely related to their base models, confirming the inheritance property. For instance, DeepSeek-distilled Qwen models locate near the Qwen3 cluster (right, purple). Third, some relationships that are not well documented appear closely in the visualization. For example, both Microsoft `orca-2-13b` and LMSYS `vicuna-7b-v1.1`, both of which are not in the "Model Tree", state in their model cards that they are fine-tuned from Llama without specifying the version. In Figure 4, `vicuna-7b-v1.1` is located within the Llama-base cluster (left-middle, green), and `orca-2-13b` lies in the Llama-chat cluster (upper-middle, green), suggesting their likely base versions in fine-tuning. This demonstrates the capability of DNA to detect related LLMs.

Table 1: DNA LLM-relation detection test performance (mean ±standard deviation across five seeds)

| Method | Accuracy | Precision | Recall | F1 | AUC |
|---|---|---|---|---|---|
| Random | 0.493 ±0.036 | 0.515 ±0.034 | 0.516 ±0.042 | 0.516 ±0.036 | 0.492 ±0.036 |
| Greedy | 0.593 ±0.018 | 0.759 ±0.081 | 0.329 ±0.000 | 0.458 ±0.014 | 0.606 ±0.023 |
| PhyloLM + SVM | 0.742 ±0.058 | 0.741 ±0.073 | 0.812 ±0.123 | 0.766 ±0.057 | 0.788 ±0.060 |
| DNA (Qwen3) + SVM | 0.919 ±0.048 | 0.898 ±0.049 | 0.957 ±0.072 | 0.925 ±0.047 | 0.979 ±0.027 |
| DNA (BGE) + SVM | **0.934 ±0.053** | 0.905 ±0.068 | **0.982 ±0.059** | **0.940 ±0.050** | **0.992 ±0.023** |
| DNA (MPNet) + SVM | 0.932 ±0.048 | **0.914 ±0.065** | 0.966 ±0.069 | 0.937 ±0.047 | 0.990 ±0.020 |

Table 2: DNA-based routing accuracy on test set

| Random | Single Best | EmbedLLM (Matrix Factorisation) | DNA |
|---|---|---|---|
| 0.402 ±0.016 | 0.607 ±0.000 | 0.665 ±0.003 | **0.672 ±0.008** |

## 5.3 MODEL ROUTING

This section investigates how DNA can benefit model routing. We use the same dataset, data split, baselines, and metric as EmbedLLM (Zhuang et al., 2025). EmbedLLM learns a compact LLM representation during routing training. To compare the quality of DNA with EmbedLLM-learned embeddings, within the matrix factorization framework of EmbedLLM, we replace the LLM embedding layer and its subsequent linear layer with frozen DNAs. These DNAs are directly compared

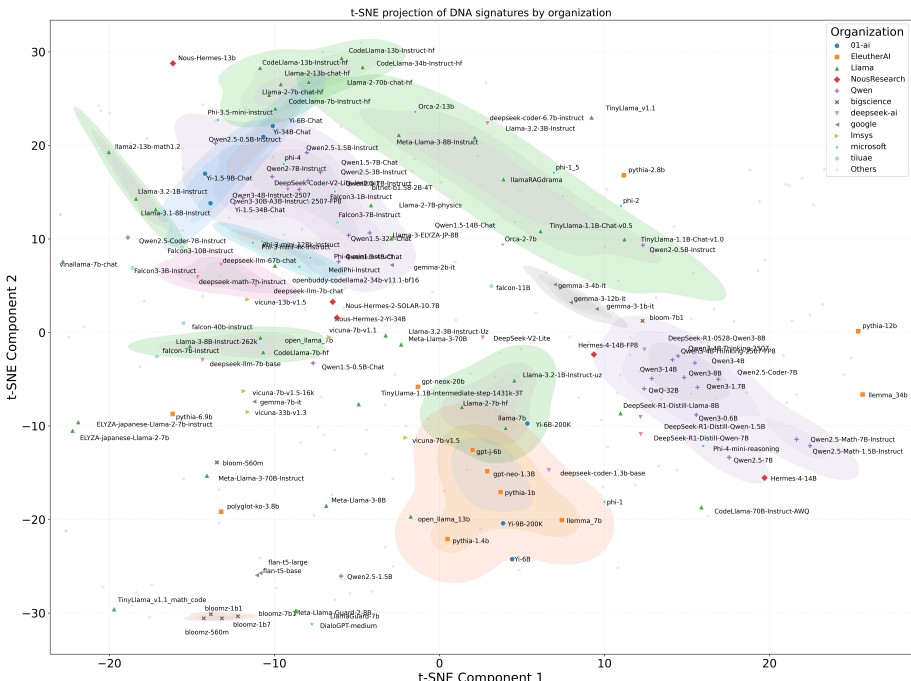

Figure 4: Visualization of DNAs by t-SNE. Colors denote organizations releasing LLMs. Organizations with fewer than five LLMs are collapsed into "Others". Background regions are obtained by localized DBSCAN started where each organization forms a group of more than three models.

with the query embeddings to make routing decisions. Notably, 11 models reported by Zhuang et al. (2025) either no longer exist or failed to run; their results are therefore omitted. Hence, our numbers do not perfectly align with those in Zhuang et al. (2025).

Table 2 reports *routing accuracy*, defined by Zhuang et al. (2025) as the fraction of queries routed to an LLM that answers correctly. The results show that frozen DNA outperforms the embeddings learned by EmbedLLM. This result is notable because EmbedLLM explicitly learns representations on this routing dataset, whereas LLM DNA is training-free and task-agnostic. Achieving parity without task-specific optimization validates the quality of DNA and highlights its superior scalability: unlike EmbedLLM, our model supports arbitrary new models without retraining.

## 5.4 STABILITY OF DNA ON HETEROGENEOUS DATASETS

This subsection examines how the choice of sampled distribution $S_t$ influences the quality of the generated DNAs. We assess this effect with a Mantel test between DNAs extracted from two disjoint datasets. Specifically, we construct one dataset as the union of 100 samples from SQuAD and 100 samples from CommonsenseQA (squad_cqa) and another as the union of 100 samples from HellaSwag and 100 samples from Winogrande (hs_wg). We compute DNAs from the same set of models, then calculate the distances between randomly selected pairs of models, visualize them in Figure 3, and compute the Pearson correlation and its $p$-value. From Figure 3, we observe a strong correlation (Pearson $r > 0.75$) between the two distance structures with very high statistical significance ($p < 0.001$). These results indicate that the structure of DNAs does not depend on the sampled data from which it is extracted, indicating the stability of our DNA extraction approach.

## 5.5 DNA EXTRACTED FROM RANDOM DATA

This subsection shows that LLM DNA remains discriminative even under random inputs. We generate 600 synthetic 100-word prompts using wonderwords.RandomWord (6×100, matching the main protocol), run each LLM to obtain outputs, embed them with Qwen3-8B-Embedding using the same DNA extraction pipeline, and evaluate DNAs via relation prediction. An exam-

ple slice of the random inputs is: "`disdain chapel intention gymnast activation ... codpiece hypothesis endothelium masonry`".

The results are reported in Table 3. Interestingly, the random-data setting even slightly outperforms the standard benchmark. This highlights that the DNA extraction method is largely insensitive to input-dataset bias. The key intuition is that DNA analyses (e.g., relationship prediction) target *relative relationships* among LLMs rather than absolute accuracy. Different datasets act as different "views" of the same model set: a suboptimal view may obscure some relations, but it is unlikely to fabricate spurious ones—for instance, it will not make two genuinely similar models appear unrelated. This observation also aligns with the stability we see under random projection.

Table 3: DNA (Qwen3) relation prediction performance (DNA extracted from random inputs)

| Method | Accuracy | Precision | Recall | F1 | AUC |
|---|---|---|---|---|---|
| PhyloLM + SVM | 0.742 ±0.058 | 0.741 ±0.073 | 0.812 ±0.123 | 0.766 ±0.057 | 0.788 ±0.060 |
| DNA + SVM | 0.919 ±0.048 | 0.898 ±0.049 | 0.957 ±0.072 | 0.925 ±0.047 | 0.979 ±0.027 |
| DNA (Rand) + SVM | **0.949 ±0.048** | **0.931 ±0.053** | **0.977 ±0.062** | **0.952 ±0.048** | **0.987 ±0.038** |

## 5.6 ABLATION STUDY FOR CHAT TEMPLATE

We study how chat templates influence DNA extraction for instruction-tuned models. Chat templates inject the special tokens and formatting used during instruction tuning to elicit conversational behavior. For an instruction-tuned model (e.g., `Llama3-Instruct`), we compare two protocols: (i) bypass the template and feed the raw prompt directly (completion-style), and (ii) apply the template and compute DNA using only the extracted assistant's response. For non-chat (base/completion) models, the extraction pipeline is unchanged. We report relationship prediction performance under the same setting as Table 4. Removing the chat template consistently improves accuracy by 1–3% across encoders, suggesting that for cross-family comparisons involving both chat-tuned and completion models, omitting templates yields a fairer and more direct functional comparison by evaluating chat-tuned models through the same completion-style interface as their base counterparts.

Table 4: DNA relation prediction performance: with vs without chat templates

| Encoder | Template | Accuracy | Precision | Recall | F1 | AUC |
|---|---|---|---|---|---|---|
| Qwen3 | w/ template | 0.919 ±0.048 | 0.898 ±0.049 | 0.957 ±0.072 | 0.925 ±0.047 | 0.979 ±0.027 |
| | w/o template | **0.930 ±0.045** | **0.916 ±0.067** | **0.961 ±0.071** | **0.935 ±0.044** | **0.994 ±0.021** |
| MPNet | w/ template | 0.932 ±0.048 | 0.914 ±0.065 | 0.966 ±0.069 | 0.937 ±0.047 | 0.990 ±0.020 |
| | w/o template | **0.958 ±0.047** | **0.943 ±0.060** | **0.982 ±0.059** | **0.960 ±0.047** | **0.994 ±0.016** |
| BGE | w/ template | 0.934 ±0.053 | 0.905 ±0.068 | **0.982 ±0.059** | 0.940 ±0.050 | **0.992 ±0.023** |
| | w/o template | **0.945 ±0.054** | **0.929 ±0.073** | 0.974 ±0.066 | **0.948 ±0.052** | **0.992 ±0.023** |

## 5.7 EFFECT OF FINE-TUNING ON THE DNA VALUES

This subsection quantifies how fine-tuning changes DNA values. Starting from `Llama3-8B-Instruct`, we fully fine-tune all parameters on subsets of the NVIDIA `OpenMathInstruct-2` dataset (Toshniwal et al., 2024) with sizes $10, 100, 1000, 10000$. Since `Llama3` was released before `OpenMathInstruct-2`, this dataset appears in neither its pretraining corpus nor our DNA extraction set. Figure 5 compares the DNAs of the original and fine-tuned models and reports their $L_2$ distances to the original DNA.

We draw two conclusions. First, the DNA distance increases with the amount of fine-tuned data, showing a monotonic but nonlinear trend. Specifically, the distance rises sharply after fine-tuning on a tiny external subset (e.g., the first 10 samples causes $L_2 = 0.69$) and then grows more gradually (up to 0.80 with 10,000 samples) as additional samples are learned. Second, the global DNA structure remains stable: most subsequences preserve their dark/light patterns with only minor variation, suggesting that fine-tuning alters local traits while leaving inherited, intrinsic behaviors largely intact.

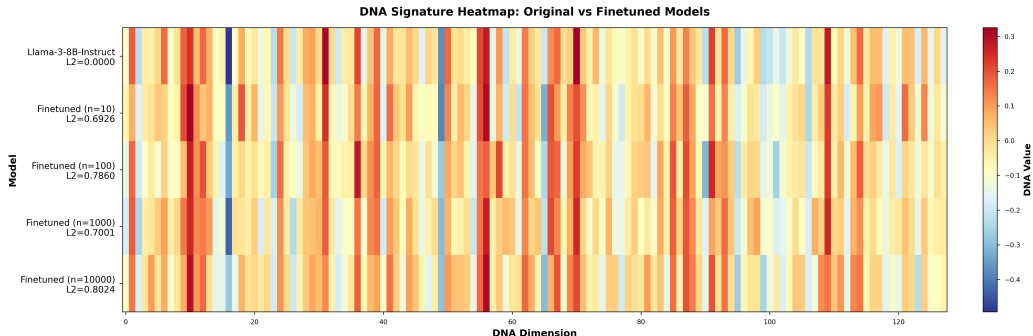

Figure 5: Shift of DNA values when (full-parameter) fine-tuning `Llama3-8B-Instruct` with `OpenMathInstruct-2` subsets of size $n$

## 5.8 PHYLOGENETIC TREE OF LLM

Analogous to phylogenetic analysis in biology, once LLM DNA is available, we can construct a phylogenetic tree to visualize model evolution. Specifically, we adopt the widely used *Neighbor-Joining (NJ)* method (Saitou & Nei, 1987) based on DNA distances, followed by the default midpoint-rooting strategy. Figure 6 visualizes this tree, with branches ordered by their leaf counts. Notably, without access to release dates, LLM DNA distances generally recover the actual evolution path, including: (i) architectural shifts from encoder-decoder (e.g., Flan-T5) to widely used decoder-only models (e.g., Llama, Qwen); (ii) temporal progression from 2023 to 2025; and (iii) lineage evolution within LLM families (e.g., Llama 2 to Llama 3). Moreover, since longer branches represent a greater functional distance between successive model families, the tree suggests distinct evolutionary speeds across model families; for instance, Qwen and Gemma evolve faster than the Llama series. A more detailed phylogenetic tree with one model per leaf is provided in Appendix C.1.

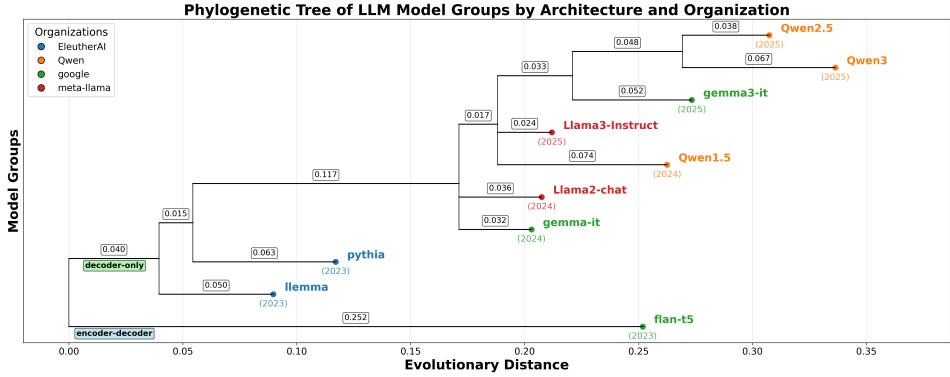

Figure 6: Phylogenetic Tree of LLM families built from DNA $\ell_2$ distances with NJ algorithm

## 6 CONCLUSION

In this paper, we introduced LLM DNA, a formal framework for representing the functional behavior of large language models. We mathematically defined LLM DNA as a low-dimensional vector, proved its existence, and established its key properties of inheritance and genetic determination. Based on this theory, we developed a practical, training-free pipeline to extract DNA from a wide range of models. Our experiments on 305 LLMs demonstrate that DNA can effectively detect both documented and undocumented evolutionary relationships, and constructing a phylogenetic tree from these DNAs provides a novel way to visualize the evolution of the LLM ecosystem.

## REPRODUCIBILITY STATEMENT

To facilitate reproducibility, we release our source code at `https://github.com/Xtra-Computing/LLM-DNA` and an interactive demo at `https://dna.xtra.science`. We also provide a PyPI package, available at `https://pypi.org/project/llm-dna`, which can be installed via `pip install llm-dna`. Regarding theoretical and experimental details, proofs appear in Appendix A, data specifications in Section 5.1, and hyperparameters, model details, and licenses in Appendix B.

## ETHICAL STATEMENT

This study does not involve human subjects. All models used are open source, and we comply with their licenses; our use is solely for research purposes. The technique does not pose any significant privacy or security risk.

## ACKNOWLEDGEMENT

This research/project is supported by the National Research Foundation, Singapore and Infocomm Media Development Authority under its Trust Tech Funding Initiative. Any opinions, findings and conclusions or recommendations expressed in this material are those of the author(s) and do not reflect the views of National Research Foundation, Singapore and Infocomm Media Development Authority.

Zhaomin Wu is also partially supported by a National Research Foundation (NRF) Postdoctoral Award.

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

# Appendix

## Table of Contents

## A  PROOF

### A.1  LLM PROPERTIES: INHERITANCE AND GENETIC DETERMINISM

Before discussing the properties, we begin by defining the evolution of an LLM.

**Definition A.1** (Evolution)**.** *Let $\mathcal{F}$ be the LLM function space in Definition 3.1, and let $d_H$ be the Hilbert distance on $\mathcal{F}$. For a threshold $\delta_H > 0$, an **evolution** is a function $\mathcal{E} : \mathcal{F} \to \mathcal{F}$ such that for any $f \in \mathcal{F}$,*

$$d_H\big(f, \mathcal{E}(f)\big) < \delta_H.$$

*If $f_2 = \mathcal{E}(f_1)$, we say $f_2$ is evolved from $f_1$ within $\delta_H$.*

This notion covers a wide range of operations that induce a bounded shift in an LLM's functional behavior, including but not limited to fine-tuning, distillation, and reinforcement learning, as they modify the LLM within a bounded Hilbert-distance threshold.

The definition of LLM DNA gives rise to several properties analogous to biological DNA. The two most fundamental are **inheritance** and **genetic determinism** introduced at the start of this section. Drawing a parallel with biology, inheritance suggests that an evolutionary process acting on an LLM, such as fine-tuning, produces descendants with similar DNAs. Genetic determinism indicates LLMs with similar DNAs behave similarly. Both properties can be mathematically derived from the definition of DNA (Definition 3.2) in following forms.

**Theorem A.2** (Inheritance). *Let an LLM $f_2 \in \mathcal{F}$ be derived from an LLM $f_1 \in \mathcal{F}$, with corresponding DNAs $\tau_{f_2}$ and $\tau_{f_1}$. For any desired DNA proximity $\epsilon_\tau > 0$, there exists a Hilbert distance threshold $\delta_H > 0$ such that if $d_H(f_1, f_2) < \delta_H$, then $d_\tau(\tau_{f_1}, \tau_{f_2}) < \epsilon_\tau$.*

*Proof.* This property is a direct consequence of the bi-Lipschitz condition in Definition 3.2. We must show that for any $\epsilon_\tau > 0$, we can find a $\delta_H > 0$ satisfying the implication:

$$d_H(f_1, f_2) < \delta_H \implies d_\tau(\tau_{f_1}, \tau_{f_2}) < \epsilon_\tau \tag{1}$$

The bi-Lipschitz condition provides the upper bound $d_\tau(\tau_{f_1}, \tau_{f_2}) \leq c_2 \cdot d_H(f_1, f_2)$ for some constant $c_2 > 0$. Let us choose $\delta_H = \epsilon_\tau / c_2$. Since $\epsilon_\tau > 0$ and $c_2 > 0$, our threshold $\delta_H$ is also positive. Now, assume $d_H(f_1, f_2) < \delta_H$ for some pair of LLMs. It follows that:

$$\begin{aligned} d_\tau(\tau_{f_1}, \tau_{f_2}) &\leq c_2 \cdot d_H(f_1, f_2) && \text{(by the Lipschitz condition)} \\ &< c_2 \cdot \delta_H && \text{(by assumption)} \\ &= c_2 \cdot \left(\frac{\epsilon_\tau}{c_2}\right) = \epsilon_\tau \end{aligned}$$

This confirms the implication in Equation 1, and the theorem holds. $\square$

**Theorem A.3** (Genetic Determinism). *Let $f_1, f_2 \in \mathcal{F}$ be two LLMs with corresponding DNAs $\tau_{f_1}, \tau_{f_2} \in \mathcal{D}$. For any desired functional proximity $\epsilon_H > 0$, there exists a DNA distance threshold $\delta_\tau > 0$ such that if $d_\tau(\tau_{f_1}, \tau_{f_2}) < \delta_\tau$, then $d_H(f_1, f_2) < \epsilon_H$.*

*Proof.* This property follows from the lower bound of the bi-Lipschitz condition. We need to show that for any given $\epsilon_H > 0$, there is a $\delta_\tau > 0$ such that:

$$d_\tau(\tau_{f_1}, \tau_{f_2}) < \delta_\tau \implies d_H(f_1, f_2) < \epsilon_H \tag{2}$$

The bi-Lipschitz condition states that $c_1 \cdot d_H(f_1, f_2) \leq d_\tau(\tau_{f_1}, \tau_{f_2})$ for some constant $c_1 > 0$. Rearranging this gives a bound on the Hilbert distance:

$$d_H(f_1, f_2) \leq \frac{1}{c_1} d_\tau(\tau_{f_1}, \tau_{f_2}) \tag{3}$$

Let us choose $\delta_\tau = \epsilon_H \cdot c_1$. Since $\epsilon_H > 0$ and $c_1 > 0$, it follows that $\delta_\tau > 0$. Assuming $d_\tau(\tau_{f_1}, \tau_{f_2}) < \delta_\tau$, we have:

$$\begin{aligned} d_H(f_1, f_2) &\leq \frac{1}{c_1} d_\tau(\tau_{f_1}, \tau_{f_2}) && \text{(from Equation 3)} \\ &< \frac{1}{c_1} \delta_\tau && \text{(by assumption)} \\ &= \frac{1}{c_1}(\epsilon_H \cdot c_1) = \epsilon_H \end{aligned}$$

Thus, the implication holds, proving the theorem. $\square$

## A.2 LLM Functional Space is a Hilbert Space

**Lemma A.4** (LLM Functional Space is a Hilbert Space). *Let $\mathcal{F}$ be the space of LLM functions mapping from the finite input space $\mathcal{S}_m = \{x_1, \ldots, x_N\}$ to the vector space of outputs $\mathcal{O} := \mathbb{R}^N$. Let $\mu : \mathcal{S}_m \to [0, 1]$ be a probability distribution on the input space, assigning a probability $\mu(x_i)$ to each sequence such that $\sum_{i=1}^{N} \mu(x_i) = 1$. The pair $(\mathcal{F}, d_H)$ forms a Hilbert space, where the distance metric $d_H$ is defined as: $d_H(f_1, f_2) = \sqrt{\sum_{i=1}^{N} \mu(x_i) \|f_1(x_i) - f_2(x_i)\|_2^2}$*

*Proof.* We prove this by showing that $\mathcal{F}$ is a vector space equipped with a valid inner product.

**Vector Space Structure.** Any function $f \in \mathcal{F}$ is uniquely determined by the tuple of its outputs. We can represent $f$ by its concatenated output vector $\mathbf{v}_f := (f(x_1), \ldots, f(x_N)) \in \mathcal{O}^N$. This establishes a natural isomorphism between $\mathcal{F}$ and the vector space $\mathcal{O}^N$, from which $\mathcal{F}$ inherits its vector space structure.

**Inner Product Validation.** We define a weighted inner product on $\mathcal{F}$ as:

$$\langle f_1, f_2 \rangle_H := \sum_{i=1}^{N} \mu(x_i) \langle f_1(x_i), f_2(x_i) \rangle_{\mathcal{O}} \tag{4}$$

where $\langle \cdot, \cdot \rangle_{\mathcal{O}}$ is the standard dot product on $\mathcal{O} = \mathbb{R}^N$. This function inherits linearity and symmetry from the standard dot product. Positive-definiteness is guaranteed by the condition that $\mu(x_i) > 0$ for all $i$. The distance metric induced by this inner product is precisely $d_H$:

$$d_H(f_1, f_2) := \sqrt{\langle f_1 - f_2, f_1 - f_2 \rangle_H} = \sqrt{\sum_{i=1}^{N} \mu(x_i) \| f_1(x_i) - f_2(x_i) \|_2^2} \tag{5}$$

As a complete inner product space, $(\mathcal{F}, d_H)$ is a Hilbert space. $\qquad\square$

### A.3 PROOF OF THEOREM 3.3

**Theorem 3.3** (Existence of LLM DNA). *For any finite set of $K$ LLMs $\mathcal{F}_K = \{f_1, \ldots, f_K\} \subset \mathcal{F}$, a DNA representation (Definition 3.2) with DNA space $\mathcal{D} = \mathbb{R}^L$ exists, satisfying for all $f_i, f_j \in \mathcal{F}_K$, $c_1 \cdot d_H(f_i, f_j) \leq d_\tau(\tau_{f_i}, \tau_{f_j}) \leq c_2 \cdot d_H(f_i, f_j)$, where the target dimension $L$ is $L = O\left( \left[ (c_2 + c_1)/(c_2 - c_1) \right]^2 \log K \right)$.*

*Proof.* We aim to construct a mapping from the LLM function space $\mathcal{F}$ to a low-dimensional DNA space $\mathcal{D}$ that satisfies the bi-Lipschitz condition in Definition 3.2 for any given constants $c_2 > c_1 > 0$. This is achieved by applying the Johnson-Lindenstrauss (JL) Lemma.

First, we define a symmetric distortion parameter $\epsilon$ and a scaling factor $\alpha$ from our target constants $c_1$ and $c_2$:

$$\epsilon = \frac{c_2 - c_1}{c_2 + c_1}, \quad \alpha = \frac{c_1 + c_2}{2} \tag{6}$$

Since $c_2 > c_1 > 0$, it follows that $\epsilon \in (0, 1)$, which is a valid distortion parameter for the JL Lemma.

We invoke the JL Lemma with this $\epsilon$. The lemma applies because, as shown in Lemma A.4, $(\mathcal{F}, d_H)$ is a Hilbert space, even when $d_H$ is defined with the weighting function $\mu(x)$. The lemma guarantees the existence of a linear map $E : \mathcal{F} \to \mathbb{R}^L$, with dimension $L = O(\epsilon^{-2} \log K)$, that embeds the vector representations of the functions. For any finite set $\{f_1, \ldots, f_K\} \subset \mathcal{F}$, the following holds for any pair $f_i, f_j$:

$$(1 - \epsilon) d_H(f_i, f_j) \leq \| E(f_i) - E(f_j) \|_2 \leq (1 + \epsilon) d_H(f_i, f_j) \tag{7}$$

Now, we define our final DNA mapping by scaling the output of $E$. The DNA of an LLM $f$ is $\tau_f := \alpha \cdot E(f)$. The distance in the DNA space is therefore $d_\tau(\tau_{f_i}, \tau_{f_j}) = \| \tau_{f_i} - \tau_{f_j} \|_2 = \| \alpha E(f_i) - \alpha E(f_j) \|_2 = \alpha \| E(f_i) - E(f_j) \|_2$.

By multiplying the inequality in Equation 7 by our scaling factor $\alpha > 0$, we get:

$$\alpha(1 - \epsilon) d_H(f_i, f_j) \leq \alpha \| E(f_i) - E(f_j) \|_2 \leq \alpha(1 + \epsilon) d_H(f_i, f_j)$$

We can now check the bounds. For the lower bound:

$$\alpha(1 - \epsilon) = \left( \frac{c_1 + c_2}{2} \right) \left( 1 - \frac{c_2 - c_1}{c_2 + c_1} \right) = \left( \frac{c_1 + c_2}{2} \right) \left( \frac{c_2 + c_1 - c_2 + c_1}{c_2 + c_1} \right) = \frac{2c_1}{2} = c_1$$

For the upper bound:

$$\alpha(1 + \epsilon) = \left( \frac{c_1 + c_2}{2} \right) \left( 1 + \frac{c_2 - c_1}{c_2 + c_1} \right) = \left( \frac{c_1 + c_2}{2} \right) \left( \frac{c_2 + c_1 + c_2 - c_1}{c_2 + c_1} \right) = \frac{2c_2}{2} = c_2$$

Substituting these results and the definition of $d_\tau$ into our inequality, we obtain:

$$c_1 \cdot d_H(f_i, f_j) \leq d_\tau(\tau_{f_i}, \tau_{f_j}) \leq c_2 \cdot d_H(f_i, f_j)$$

This explicitly provides the required inequality. Thus, a valid DNA mapping exists for any choice of constants $c_2 > c_1 > 0$. $\qquad\square$

## A.4 PROOF OF THEOREM 4.2

**Lemma 4.2** (Concentration of Empirical Functional Distance). *Let $f_1, f_2 \in \mathcal{F}$ be two LLM functions. Let $\hat{d}_f^2 = \sum_{j=1}^t \|f_1(x_j) - f_2(x_j)\|_2^2$ be the squared empirical functional distance calculated from a sample of $t$ i.i.d. inputs $\{x_1, \ldots, x_t\}$. Let $d_H^2$ be the true squared Hilbert distance. Assume the squared Euclidean distance between any two output vectors is bounded such that $\|f_1(x) - f_2(x)\|_2^2 \le C_{\max}$ for any input $x$ and for all functions of interest. Then, for any $\epsilon > 0$, the following bound holds for the sample average:* $P\left(\left|\frac{1}{t}\hat{d}_f^2 - d_H^2\right| \ge \epsilon\right) \le 2\exp\left(-\frac{2t\epsilon^2}{C_{\max}^2}\right).$

*Proof.* Let the sample of $t$ inputs, drawn i.i.d. from the distribution $\mu$, be $\{x_1, \ldots, x_t\}$. Let $Z_1, \ldots, Z_t$ be a sequence of i.i.d. random variables, where each variable $Z_j$ represents the squared Euclidean distance between the output logit vectors for a randomly drawn input $x_j$:

$$Z_j := \|f_1(x_j) - f_2(x_j)\|_2^2$$

By the assumption, each $Z_j$ is bounded within the range $[0, C_{\max}]$. The true squared Hilbert distance is, by definition, the expectation of this random variable:

$$d_H^2 = \mathbb{E}[Z_j]$$

The sample mean of these $t$ variables is precisely the averaged squared empirical functional distance:

$$\frac{1}{t}\hat{d}_f^2 = \frac{1}{t}\sum_{j=1}^t Z_j$$

Hoeffding's inequality provides a bound on the probability that a sample mean of bounded, independent random variables deviates from its expected value. For a set of i.i.d. variables $\{Y_1, \ldots, Y_t\}$ bounded in $[a, b]$, the inequality states:

$$P\left(\left|\frac{1}{t}\sum_{j=1}^t Y_j - \mathbb{E}[Y]\right| \ge \epsilon\right) \le 2\exp\left(-\frac{2t\epsilon^2}{(b-a)^2}\right)$$

We apply this inequality to our variables $Z_j$. In our context, the sample mean is $\frac{1}{t}\hat{d}_f^2$, the expectation is $d_H^2$, and the range $[a, b]$ is $[0, C_{\max}]$, making $(b-a)^2 = C_{\max}^2$. Substituting these into the general form directly yields the desired result:

$$P\left(\left|\frac{1}{t}\hat{d}_f^2 - d_H^2\right| \ge \epsilon\right) \le 2\exp\left(-\frac{2t\epsilon^2}{C_{\max}^2}\right)$$

This completes the proof. $\qquad\square$

## A.5 PROOF OF COROLLARY 3.4

**Corollary 3.4** (Construct LLM DNA via Random Projection). *For any finite set of $K$ LLMs $\mathcal{F}_K = \{f_1, \ldots, f_K\} \subset \mathcal{F}$, the LLM DNA in Theorem 3.3 can be obtained from a random linear projection $E : \mathcal{F} \to \mathbb{R}^L$ with probability at least $1 - 2/K^2$.*

*Proof.* Since Lemma A.4 shows that $\mathcal{F}$ is a Hilbert space, the JL lemma applies. Invoking a standard formulation of the lemma (e.g., Theorem 2.1 in Dasgupta & Gupta (2003)) for the set of vector representations of LLMs in $\mathcal{F}_K$ yields the desired projection and associated probability bound. $\quad\square$

## B EXPERIMENTAL DETAILS

### B.1 HYPERPARAMETERS

Unless otherwise specified, we adopt the following default configuration. The DNA dimensionality is reduced to 128 using Gaussian Random Projection by default. Text embeddings are extracted with the state-of-the-art open-source encoder `Qwen/Qwen3-Embedding-8B`, using a maximum

input length of 1024 tokens with `cls` pooling. This model is also used to encode questions in routing experiment for semantic consistency. For text generation, we set `max_length` = 1024, `temperature` = 0.7, `top_p` = 0.9, if these options are applicable. Models with fewer than 7B parameters are run in BF16/FP16 precision without quantization, while larger models are automatically quantized to 8-bit for memory efficiency. For instruction-tuned models, the provided chat template is applied and only the responses are used for DNA extraction. All other parameters follow HuggingFace defaults. All applicable experiments are conducted on five seeds with mean and standard deviation reported.

## B.2 Environment

DNA extraction for each model is conducted on a single GPU. Models with up to 30B parameters are executed on NVIDIA A100 or H100 GPUs with 80GB memory, while models between 30B and 70B parameters are deployed on H200 GPUs with 140GB memory. In total, the storage footprint of the 305 models amounts to approximately 20 TB.

## C Supplementary Experiments

### C.1 Phylogenetic Tree of LLMs

Analogous to phylogenetic research in biology, once the DNA of LLMs is available, we can construct a phylogenetic tree to visualize how LLMs evolve. Unlike existing phylogenetic trees based on release histories or documentation, our phylogenetic tree is derived directly from model behavior. This enables us to uncover latent relationships between models, including close connections that may not have been explicitly documented.

To construct the phylogenetic tree, we adopt the widely used *Neighbor-Joining (NJ)* method from biology (Saitou & Nei, 1987). The core assumption of this method is to find a tree that minimizes the total "evolutionary effort," quantified as the sum of edge lengths. The evolutionary root is estimated by the midpoint of the longest path in the tree. We apply this method to LLM DNA by treating each LLM as a species and using Euclidean distance between DNA vectors as the pairwise distance measure. As an expanded version of Figure 6, the full phylogenetic tree is shown in Figure 7, from which several interesting observations emerge:

1. **LLMs from the same family are consistently colocated** (e.g., Google Flan models, Qwen3 models, Llama 3 models), confirming that the DNA representation captures series-level similarity.
2. **The DNA-based model tree aligns with the temporal evolution of LLMs**, progressing (i) from Gemma to Gemma 3 along root-to-leaf paths and (ii) from early encoder–decoder architectures (e.g., Flan-T5) to prevalent decoder-only architectures. This indicates that DNA is an effective tool for studying LLM evolution.

### C.2 Detailed Analysis of Falsely Predicted LLM Relationship

To investigate the causes of DNA false positives, we manually examined these cases with notable examples listed in Table 5. We found that a substantial portion involve pairs absent from the Hugging Face model tree yet verifiably related based on public documentation, or belonging to the same families and thus likely related. The pairs listed in Table 5 confirm that many "false positives" arise from incomplete labeling of the model-tree data rather than from the DNA method itself. Moreover, LLM DNA offers a principled way to enrich the model tree.

### C.3 Ablation Study of Bottleneck Embedding Model

This ablation study evaluates the effect of the bottleneck embedding model on the computed DNA. In addition to Qwen-8B-Embedding, we compute DNA using two popular sentence-embedding models from different organizations and with different sizes: `all-mpnet-base-v2` (0.1B parameters) and `bge-large-en-v1.5` (0.3B parameters). We then perform a Mantel test—a standard statistical test for correlating two distance matrices—pairwise across the three DNA variants. Specifically,

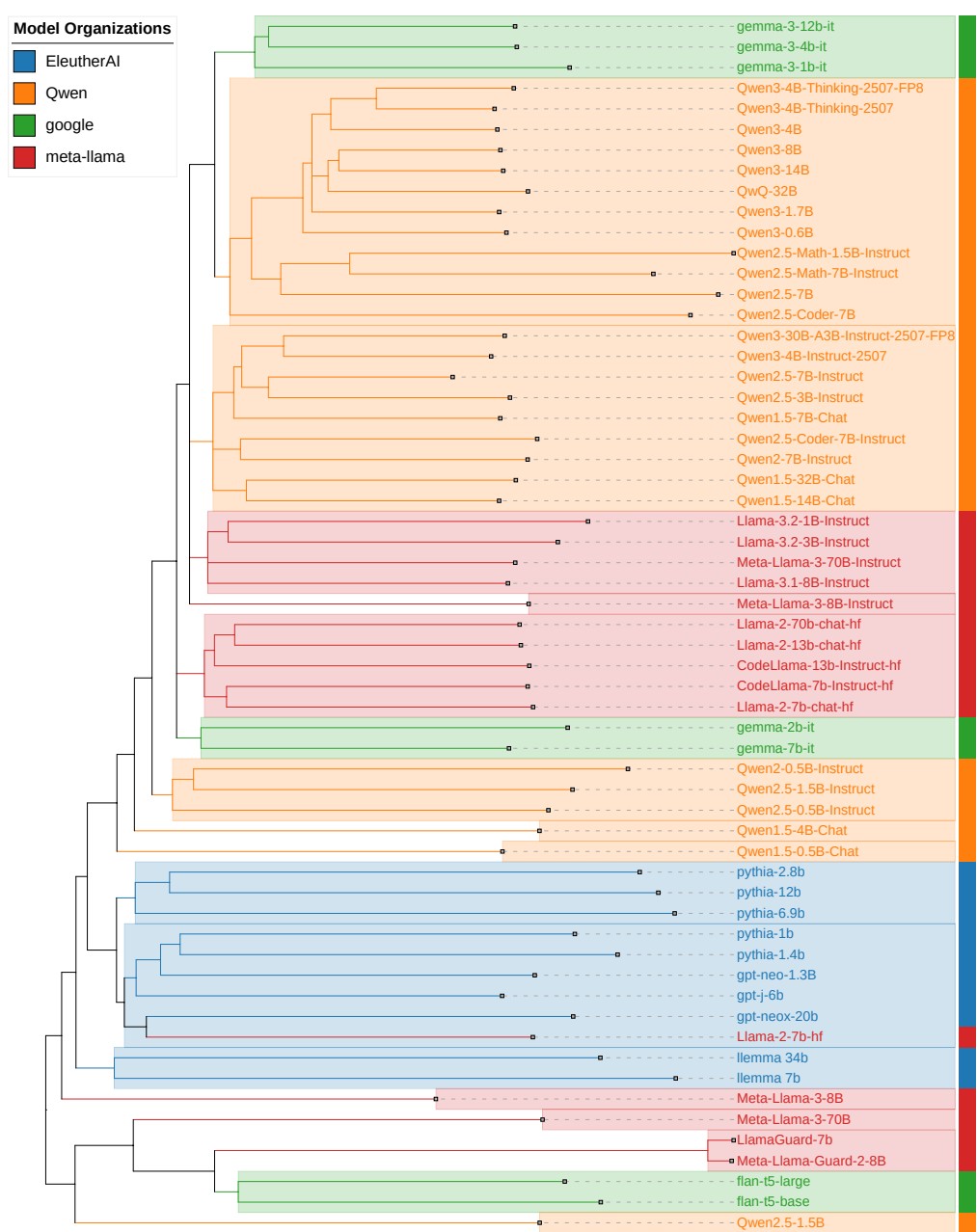

Figure 7: Phylogenetic tree of representative LLMs constructed using the Neighbor-Joining algorithm on their DNA embeddings.

we sample 10,000 random DNA pairs, compute their distances, and report the Spearman correlation and p-value in Figure 8. **All pairs of embedding model exhibit strong agreement, with high correlations ($r > 0.75$) and strong statistical significance ($p < 0.0001$).** These results indicate that the choice of embedding model has marginal impact on the overall DNA structure; even a small (0.1B) embedding model yields comparable DNA.

## C.4 Ablation Study of the Embedding Model's Output Dimension $p$

This ablation study investigates the effect of the pre-aggregation embedding dimension $p$ from Qwen3-8B-Embedding on DNA quality. We vary $p \in 4, 64, 128, 1024, 4096$ and evaluate each setting on the same relationship prediction task used in Table 1. The results are shown in Figure 9.

Table 5: Examples of Undocumented Relationships Identified by LLM DNA

| Model 1 | Model 2 |
|---|---|
| Qwen/Qwen2.5-7B-Instruct | Qwen/Qwen3-1.7B |
| Qwen/Qwen2.5-7B-Instruct | Qwen/Qwen3-14B |
| Qwen/Qwen2.5-7B-Instruct | Qwen/Qwen3-8B |
| Qwen/Qwen2.5-Coder-7B-Instruct | Qwen/Qwen3-4B-Thinking-2507-FP8 |
| Qwen/Qwen3-4B | Qwen/Qwen2.5-Coder-7B |
| Qwen/Qwen3-4B | Qwen/Qwen3-8B |
| Qwen/Qwen3-4B-Thinking-2507-FP8 | Qwen/Qwen2.5-0.5B-Instruct |
| Qwen/Qwen3-4B-Thinking-2507-FP8 | Qwen/Qwen2.5-7B |
| Qwen/Qwen3-8B | Qwen/Qwen3-4B-Thinking-2507-FP8 |
| prithivMLmods/rStar-Coder-Qwen3-0.6B | Qwen/Qwen2.5-Coder-7B |
| prithivMLmods/rStar-Coder-Qwen3-0.6B | janhq/Jan-v1-4B |
| prithivMLmods/rStar-Coder-Qwen3-0.6B | meta-llama/Meta-Llama-3-8B-Instruct |
| prithivMLmods/rStar-Coder-Qwen3-0.6B | openai/gpt-oss-20b |
| tiiuae/falcon-7b-instruct | Qwen/Qwen2.5-Coder-7B |
| tomg-group-umd/DynaGuard-8B | Qwen/Qwen3-14B |

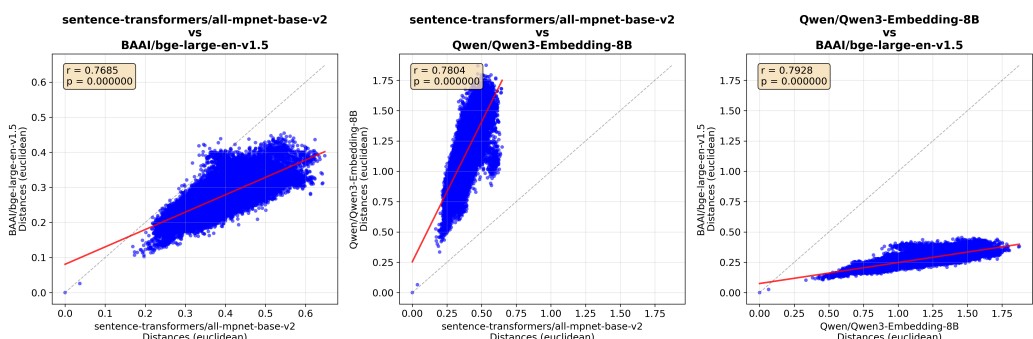

Figure 8: Mantel test result across three different bottleneck sentence-embedding models

Across all metrics, performance remains largely unchanged, indicating that $p$ has only a marginal impact. This suggests that **DNA quality is not primarily driven by the embedding model's output dimension, but by the subsequent aggregation process**. It also reinforces the observation in Table 1 that even simple sentence-embedding models can achieve strong relation prediction performance.

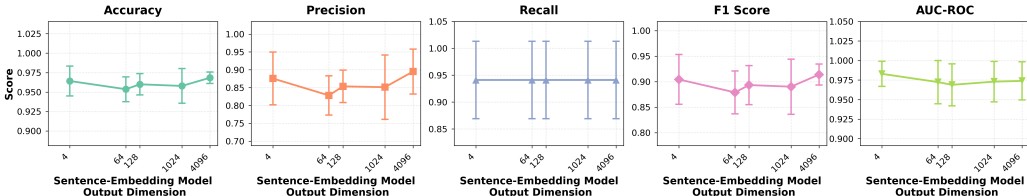

Figure 9: Relationship prediction performance (Table 1) under different output dimensions $p$ of the sentence-embedding model. Results are averaged over five random seeds; error bars denote standard deviation.

## C.5 Ablation Study of the DNA Dimension $L$

This ablation study examines how the DNA dimension $L$ affects DNA quality. We vary $L$ in $4, 32, 128, 1024, 4096$ and evaluate each setting on the same relationship prediction task used in Table 1. The results are shown in Figure 10. **DNA quality improves as $L$ increases and converges around $L = 128$.** Very low dimensions (e.g., $L = 4$) underperform because they discard structural information from the original high-dimensional space. Thus, selecting $L$ involves a **tradeoff**

**between quality and efficiency**: larger dimensions preserve more structural detail and improve accuracy, but incurs increasing downstream computation and storage costs; smaller dimensions are cheaper and faster, but can overly compress the space and lose critical structure, leading to degraded DNA quality. We therefore use $L = 128$ in all experiments. Note that this choice is specific to our current evaluation scale of approximately 300 LLMs; for substantially larger model sets, a larger $L$ may be required.

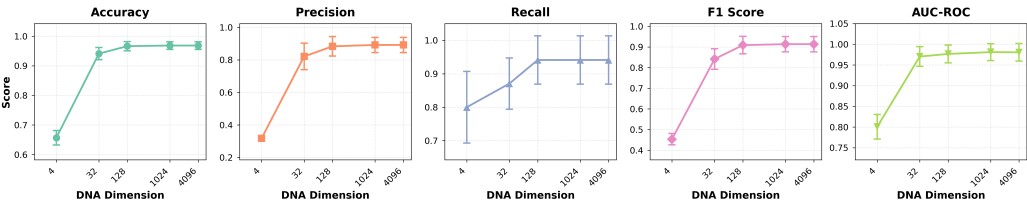

Figure 10: Relationship prediction performance (Table 1) under different DNA dimensions $L$. Results are averaged over five random seeds; error bars denote standard deviation.

### C.6 EFFECT OF LLM'S PARAMETER SIZE ON DNA QUALITY

This subsection investigates whether the performance of LLM DNA is biased by model scale—specifically, whether the method effectiveness varies significantly between small and large models. To rigorously test this, we conduct a post-hoc error analysis completely separate from the DNA classification pipeline. The DNA classifier first predicts relationships using only functional embeddings, without access to metadata such as parameter counts. After fixing these predictions, we calculate the *cumulative parameter size* for every test pair (the sum of parameters of both models). We then compute the *Point-Biserial correlation ($r_{pb}$)* and fit an independent logistic regression model to determine if this cumulative size can predict the *correctness* of the DNA classifier's output (1 for correct, 0 for incorrect). Table 6 reports the performance metrics alongside the correlation coefficient and the $p$-value of the size coefficient derived from the regression analysis.

The results yield two key observations. First, **DNA quality is statistically independent of LLM size**: across all DNA-based methods, the Point-Biserial correlations are negligible (ranging from -0.006 to 0.023), and the $p$-values for the size coefficient are substantial (ranging from 0.63 to 0.69), well above the standard significance threshold of 0.05. This indicates that the method's ability to detect evolutionary relationships does not degrade for larger models nor improve for smaller ones. Second, **robustness is consistent across different encoders regardless of their own size**: the lack of size bias (high $p$-value) remains constant across the large Qwen3 (8B) encoder and the much smaller BGE (0.3B) and MPNet (0.1B) encoders. This suggests that LLM DNA captures intrinsic functional behaviors that persist across model scales, rather than relying on complexity metrics associated with parameter count.

Table 6: Point-Biserial correlation ($r_{pb}$) and statistical significance ($p$-value) between LLM size and relation prediction accuracy

| Method | Accuracy | F1 | AUC | Corr. ($r_{pb}$) | $p$-value (Size) |
|---|---|---|---|---|---|
| PhyloLM | 0.742 ±0.058 | 0.766 ±0.057 | 0.788 ±0.060 | -0.097 ±0.195 | 0.5202 |
| DNA (Qwen3) | 0.919 ±0.048 | 0.925 ±0.047 | 0.979 ±0.027 | 0.025 ±0.123 | 0.6349 |
| DNA (BGE) | 0.934 ±0.053 | 0.940 ±0.050 | 0.992 ±0.023 | 0.013 ±0.125 | 0.6550 |
| DNA (MPNet) | 0.932 ±0.048 | 0.937 ±0.047 | 0.990 ±0.020 | 0.023 ±0.122 | 0.6903 |

### C.7 RELATION PREDICTION FOR LLMS WITH SAME TOKENIZER

Though our state-of-the-art baseline PhyloLM (Yax et al., 2025) is designed to compare general LLMs across both same- and different-tokenizer settings, it introduces different comparison strategies for these two cases. In particular, when models share a tokenizer, PhyloLM can directly leverage exact token distributions, which typically yields stronger performance than its cross-tokenizer variant based on the first four characters (as shown in Table 1). To provide a fair and transparent com-

parison under PhyloLM's same-tokenizer regime, we further compare PhyloLM with LLM DNA on a single-tokenizer subset. This setting is more challenging because models with the same tokenizer are typically closer in function and thus harder to distinguish. Concretely, we filter the dataset to include only models using Qwen2Tokenizer, the most commonly adopted tokenizer among our evaluated LLMs, and perform the same relation prediction task. The results are reported in Table 7.

Comparing Table 7 with Table 1 yields several key findings. First, **all methods except PhyloLM exhibit a clear performance drop in the same-tokenizer setting**: for example, DNA (MPNet) decreases from 0.93 to 0.86 in accuracy, and Greedy drops from 0.59 to 0.42. In contrast, PhyloLM slightly improves its accuracy (0.742 to 0.755) and achieves a notable precision gain (0.74 to 0.83), confirming that the same-tokenizer case is indeed more challenging while PhyloLM remains robust. Second, **despite the reduced gap, PhyloLM is still consistently outperformed by DNA**. This may be because PhyloLM primarily evaluates the distribution of the first token, which may be insufficient to capture long-range sentence-level distributions that are better reflected in DNA. We note that this comparison used Qwen2Tokenizer to align the two methods instead of using the first 4 characters which is technically a slightly different version of PhyloLM. We also emphasize that the comparison is conducted on relation prediction, a task native to LLM DNA rather than the primary focus of PhyloLM. This does not diminish PhyloLM's demonstrated effectiveness in its intended applications, such as benchmark performance prediction. In future work, to enable a fairer and more comprehensive comparison, we will evaluate both methods on the datasets and tasks originally used to develop PhyloLM, such as model performance prediction benchmarks. Furthermore, we will investigate more deeply how sentence-level, long-range patterns affect LLM representations.

Table 7: DNA relation pediction performance for LLMs with Qwen2Tokenizer

| Method | Accuracy | Precision | Recall | F1 | AUC |
|---|---|---|---|---|---|
| Random | 0.511 ±0.054 | 0.682 ±0.073 | 0.529 ±0.054 | 0.593 ±0.047 | 0.501 ±0.068 |
| Greedy | 0.428 ±0.043 | 0.687 ±0.101 | 0.294 ±0.000 | 0.410 ±0.018 | 0.500 ±0.049 |
| PhyloLM + SVM | 0.755 ±0.136 | 0.832 ±0.080 | 0.800 ±0.204 | 0.801 ±0.149 | 0.760 ±0.180 |
| DNA (Qwen3) + SVM | 0.834 ±0.092 | 0.863 ±0.073 | 0.907 ±0.123 | 0.878 ±0.072 | 0.926 ±0.105 |
| DNA (BGE) + SVM | **0.869 ±0.078** | 0.865 ±0.069 | **0.966 ±0.094** | **0.909 ±0.059** | 0.944 ±0.087 |
| DNA (MPNet) + SVM | 0.866 ±0.088 | **0.879 ±0.099** | 0.948 ±0.098 | 0.906 ±0.065 | **0.960 ±0.079** |

## D  BROADER IMPACT

The utility of the phylogenetic tree extends beyond historical visualization, offering actionable insights into three key aspects of model management.

**License Governance.**  As the number of models grows rapidly, non-compliance with model licenses is becoming a critical concern. Such behaviors include fine-tuning or distillation from a model without giving proper credit or respecting license terms. For example, it has been alleged that certain proprietary models are derived from open-source counterparts like Qwen-2.5-14B, a claim often difficult to verify through metadata alone. While existing license-analysis tools such as ModelGo primarily rely on explicit claims, our phylogenetic analysis provides a functional indication of lineage. Similar methods have been explored in other domains; for instance, RAI2 (Dong et al., 2023) develops identification techniques for Computer Vision (CV) models to accurately identify model provenance and detect unauthorized reuse. In the language domain, our tree clusters models like Microsoft's orca-2-13b and LMSYS's vicuna-7b-v1.1 tightly, identifying a shared lineage that is not explicitly documented in Hugging Face. Ultimately, the phylogenetic tree offers a data-driven framework for flagging potential license violations and strengthening governance across the AI ecosystem.

**Model Selection.**  Our framework facilitates informed model selection by analyzing the interplay between functional inheritance and iteration velocity. First, the phylogenetic lineage allows users to identify families where specialized capabilities—such as reasoning patterns (Huang & Chang, 2023) or domain knowledge (Wu et al., 2025)—are likely transferred down from ancestor models, ensuring the retention of desired baseline traits. Second, the "evolutionary rate" (branch length) serves as a

proxy for *Iteration Velocity*, quantifying the magnitude of functional change between versions. This reveals a fundamental trade-off between novelty and stability: while a high iteration velocity offers significant behavioral shifts advantageous for *experimental use cases*, models with lower velocity (higher backward compatibility) are often preferable for *production environments*—particularly multi-agent systems—to minimize the risk of breaking established agent workflows or prompt dependencies.

**Risk Mitigation.**    Finally, the phylogenetic tree facilitates proactive risk mitigation by tracing the propagation of safety risks. Our findings on inheritance suggest that vulnerabilities—such as implanted backdoors (Yang et al., 2024), jailbreak susceptibility (Russinovich et al., 2025), or deception behaviors (Wu et al., 2026)—are probably transferred down model lineages. Consequently, the tree allows auditors to propagate known risk profiles from ancestor models to their descendants. If a parent model is flagged as unsafe, its descendants identified by the tree can be immediately prioritized for safety interventions. This enables auditors to mitigate deployment risks efficiently, focusing resources on high-risk lineages without requiring exhaustive ground-up testing for every new model iteration.

## E    LIMITATIONS

This section discusses the limitations of our work.

**Traits and DNA subsequences.**    Our formal definition of LLM DNA does not assign mathematical meaning to DNA subsequences. However, in Section 5.7, we observe that fine-tuning alters only specific subsequences while leaving the overall pattern largely intact. This suggests that certain DNA subsequences may encode distinct model traits (e.g., mathematical reasoning). A theoretical characterization of how different evolutions affect such traits is an important and challenging direction for future work. In the long run, this could enable more precise understanding, management, and even targeted modification of LLM DNA, drawing a closer parallel to how biogenetics studies and manipulates inherited properties.

**Adaptive attacks on DNA.**    Our current DNA extraction is not designed to be robust against adaptive attacks. If an attacker knows the extraction data, they could deliberately train a model toward a malicious objective—e.g., producing a substantially different DNA while functionally copying another model—in order to evade lineage detection, patents, or license constraints. A straightforward mitigation is to keep extraction datasets closed-source or to re-extract DNA using a fresh dataset when needed. A more principled direction is to scale the extraction set so that overfitting to it becomes prohibitively expensive and would noticeably degrade general performance. Developing such security guarantees remains a key topic for future work.

**Data bias and contamination.**    Our DNA extraction relies on six public datasets, which may introduce evaluation bias or contamination. Bias could cause us to overlook certain differences or overestimate similarity among some model groups, while contamination could arise if parts of the extraction data were seen during pretraining. Nonetheless, their impact on our evaluation is limited. As shown in Section 5.5, DNA remains effective even when evaluated on purely random input data. This observation supports that DNA primarily captures *relative relationships* among LLMs, rather than depending on absolute accuracy on any particular dataset. One remaining challenge is that certain model APIs, such as the Claude series, may reject outputs generated from random strings; these cases require more advanced randomized methods to bypass such safety or formatting barriers.

## F    MODEL DETAILS

In this section, we present the full list of models used, including their architectures, parameter counts, and licenses. This paper focuses on text-generation models, including decoder-only models (e.g., GPT) and encoder-decoder models (e.g., T5). For completion models, the completed text is treated as the response; for instruction-tuned models, the entire response extracted from the template is used. This may include any model-generated reasoning traces (e.g., in Qwen3), which are also considered part of the response. All uses comply with the respective model licenses.

## G    LARGE LANGUAGE MODEL USAGE

Large language models assisted in verifying theoretical results, implementing source code, and polishing this paper. The theoretical results and proofs were drafted by humans and verified by Gemini-2.5-Pro. Code implementation was primarily assisted by Claude-4-Sonnet and GPT-5-Codex, and the code's correctness and alignment with the paper were checked by Gemini-2.5-Pro. Manuscript polishing was assisted by Gemini-2.5-Pro and GPT-5.

Table 8: Full list of used 305 models, including their architectures, parameter counts, and licenses

| Model | Architecture | Parameters | License |
|---|---|---|---|
| 01-ai/Yi-1.5-34B-Chat | Decoder Only | 34B | Apache-2.0 |
| 01-ai/Yi-1.5-9B-Chat | Decoder Only | 9B | Apache-2.0 |
| 01-ai/Yi-34B-Chat | Decoder Only | 34B | Apache-2.0 |
| 01-ai/Yi-6B | Decoder Only | 6B | Apache-2.0 |
| 01-ai/Yi-6B-200K | Decoder Only | 6B | Apache-2.0 |
| 01-ai/Yi-6B-Chat | Decoder Only | 6B | Apache-2.0 |
| 01-ai/Yi-9B-200K | Decoder Only | 9B | Apache-2.0 |
| AdaptLLM/finance-chat | Decoder Only | 7B | Llama-2 |
| AdaptLLM/medicine-LLM | Decoder Only | 7B | Unknown |
| AdaptLLM/medicine-LLM-13B | Decoder Only | 13B | Apache-2.0 |
| AdaptLLM/medicine-chat | Decoder Only | 7B | Llama-2 |
| Aurore-Reveil/Koto-Small-7B-IT | Decoder Only | 8B | MIT |
| BioMistral/BioMistral-7B | Decoder Only | 7B | Apache-2.0 |
| BioMistral/BioMistral-7B-DARE | Decoder Only | 7B | Apache-2.0 |
| Biomimicry-AI/ANIMA-Nectar-v2 | Decoder Only | 7B | MIT |
| ByteDance-Seed/Seed-Coder-8B-Instruct | Decoder Only | 8B | MIT |
| CausalLM/34b-beta | Decoder Only | 34B | GPL-3.0 |
| ChrisMcCormick/deepseek-tiny-v0.1 | Decoder Only | 17.1M | Apache-2.0 |
| ClosedCharacter/Peach-2.0-9B-8k-Roleplay | Decoder Only | 9B | MIT |
| ConvexAI/Luminex-34B-v0.1 | Decoder Only | 34B | Other |
| ConvexAI/Luminex-34B-v0.2 | Decoder Only | 34B | Other |
| CorticalStack/pastiche-crown-clown-7b-dare-dpo | Decoder Only | 7B | Apache-2.0 |
| CultriX/NeuralTrix-bf16 | Decoder Only | 7B | Apache-2.0 |
| DatarusAI/Datarus-R1-14B-preview | Decoder Only | 15B | Apache-2.0 |
| DavidAU/Qwen3-Horror-Instruct-Uncensored-... | Decoder Only | 4B | Apache-2.0 |
| DeepHat/DeepHat-V1-7B | Decoder Only | 8B | Apache-2.0 |
| EleutherAI/gpt-j-6b | Decoder Only | 6B | Apache-2.0 |
| EleutherAI/gpt-neo-1.3B | Decoder Only | 1B | MIT |
| EleutherAI/gpt-neox-20b | Decoder Only | 21B | Apache-2.0 |
| EleutherAI/llemma_34b | Decoder Only | 34B | Llama-2 |
| EleutherAI/llemma_7b | Decoder Only | 7B | Llama-2 |
| EleutherAI/polyglot-ko-3.8b | Decoder Only | 3.8B | Apache-2.0 |
| EleutherAI/pythia-1.4b | Decoder Only | 2B | Apache-2.0 |
| EleutherAI/pythia-12b | Decoder Only | 12B | Apache-2.0 |
| EleutherAI/pythia-1b | Decoder Only | 1B | Apache-2.0 |
| EleutherAI/pythia-2.8b | Decoder Only | 3B | Apache-2.0 |
| EleutherAI/pythia-6.9b | Decoder Only | 7B | Apache-2.0 |
| FallenMerick/MN-Violet-Lotus-12B | Decoder Only | 12B | CC-BY-4.0 |
| FelixChao/llama2-13b-math1.2 | Decoder Only | 13B | Unknown |
| FelixChao/vicuna-7B-chemical | Decoder Only | 7B | Apache-2.0 |
| FelixChao/vicuna-7B-physics | Decoder Only | 7B | Unknown |
| FlareRebellion/WeirdCompound-v1.6-24b | Decoder Only | 24B | Unknown |
| Harshvir/Llama-2-7B-physics | Decoder Only | 7B | Unknown |
| HuggingFaceH4/zephyr-7b-beta | Decoder Only | 7B | MIT |
| HuggingFaceTB/SmolLM-135M | Decoder Only | 0.1B | Apache-2.0 |
| HuggingFaceTB/SmolLM2-135M | Decoder Only | 0.1B | Apache-2.0 |
| HuggingFaceTB/SmolLM3-3B | Decoder Only | 3B | Apache-2.0 |
| Intel/neural-chat-7b-v3-3 | Decoder Only | 7B | Apache-2.0 |
| K-intelligence/Midm-2.0-Base-Instruct | Decoder Only | 12B | MIT |
| KoboldAI/GPT-Neo-2.7B-Horni-LN | Decoder Only | 2.7B | Unknown |
| KoboldAI/OPT-13B-Erebus | Decoder Only | 13B | Other |
| LGAI-EXAONE/EXAONE-4.0-1.2B | Decoder Only | 1B | Other |
| LLM360/Amber | Decoder Only | 7B | Apache-2.0 |
| LatitudeGames/Wayfarer-2-12B | Decoder Only | 12B | Apache-2.0 |
| LeoLM/leo-hessianai-13b | Decoder Only | 13B | Unknown |
| LiquidAI/LFM2-1.2B | Decoder Only | 1B | Other |
| LiquidAI/LFM2-350M | Decoder Only | 0.4B | Other |
| MaziyarPanahi/WizardLM-Math-70B-v0.1 | Decoder Only | 69B | AGPL-3.0 |
| Neko-Institute-of-Science/metharme-7b | Decoder Only | 7B | Unknown |
| Neko-Institute-of-Science/pygmalion-7b | Decoder Only | 7B | Unknown |
| Nexusflow/Starling-LM-7B-beta | Decoder Only | 7B | Apache-2.0 |
| NinedayWang/PolyCoder-2.7B | Decoder Only | 2.7B | Unknown |
| NousResearch/Hermes-4-14B | Decoder Only | 425k | Apache-2.0 |
| NousResearch/Hermes-4-14B-FP8 | Decoder Only | 15B | Apache-2.0 |
| NousResearch/Nous-Hermes-13b | Decoder Only | 13B | gpl |
| NousResearch/Nous-Hermes-2-SOLAR-10.7B | Decoder Only | 11B | Apache-2.0 |
| NousResearch/Nous-Hermes-2-Yi-34B | Decoder Only | 34B | Apache-2.0 |
| OddTheGreat/Circuitry_24B_V.2 | Decoder Only | 24B | Unknown |
| OpenAssistant/oasst-sft-4-pythia-12b-epoch-3.5 | Decoder Only | 12B | Apache-2.0 |
| OpenBuddy/openbuddy-codellama2-34b-v11.1-bf16 | Decoder Only | 34B | Unknown |
| PharMolix/BioMedGPT-LM-7B | Decoder Only | 7B | Apache-2.0 |
| Plaban81/Moe-4x7b-math-reason-code | Decoder Only | 24B | Apache-2.0 |
| PocketDoc/Dans-PersonalityEngine-V1.3.0-24b | Decoder Only | 24B | Apache-2.0 |
| PygmalionAI/pygmalion-2.7b | Decoder Only | 2.7B | creativeml-openrail-m |
| QuixiAI/WizardLM-13B-Uncensored | Decoder Only | 13B | Other |

Table 8: Full list of used 305 models (continued)

| Model | Architecture | Parameters | License |
| --- | --- | --- | --- |
| Qwen/QwQ-32B | Decoder Only | 33B | Apache-2.0 |
| Qwen/Qwen1.5-0.5B-Chat | Decoder Only | 0.6B | Other |
| Qwen/Qwen1.5-14B-Chat | Decoder Only | 14B | Other |
| Qwen/Qwen1.5-32B-Chat | Decoder Only | 33B | Other |
| Qwen/Qwen1.5-4B-Chat | Decoder Only | 4B | Other |
| Qwen/Qwen1.5-7B-Chat | Decoder Only | 8B | Other |
| Qwen/Qwen2-0.5B-Instruct | Decoder Only | 0.5B | Apache-2.0 |
| Qwen/Qwen2-7B-Instruct | Decoder Only | 8B | Apache-2.0 |
| Qwen/Qwen2.5-0.5B-Instruct | Decoder Only | 0.5B | Apache-2.0 |
| Qwen/Qwen2.5-1.5B | Decoder Only | 2B | Apache-2.0 |
| Qwen/Qwen2.5-1.5B-Instruct | Decoder Only | 2B | Apache-2.0 |
| Qwen/Qwen2.5-3B-Instruct | Decoder Only | 3B | Other |
| Qwen/Qwen2.5-7B | Decoder Only | 8B | Apache-2.0 |
| Qwen/Qwen2.5-7B-Instruct | Decoder Only | 8B | Apache-2.0 |
| Qwen/Qwen2.5-Coder-7B | Decoder Only | 8B | Apache-2.0 |
| Qwen/Qwen2.5-Coder-7B-Instruct | Decoder Only | 8B | Apache-2.0 |
| Qwen/Qwen2.5-Math-1.5B-Instruct | Decoder Only | 2B | Apache-2.0 |
| Qwen/Qwen2.5-Math-7B-Instruct | Decoder Only | 8B | Apache-2.0 |
| Qwen/Qwen3-0.6B | Decoder Only | 0.8B | Apache-2.0 |
| Qwen/Qwen3-1.7B | Decoder Only | 2B | Apache-2.0 |
| Qwen/Qwen3-14B | Decoder Only | 15B | Apache-2.0 |
| Qwen/Qwen3-30B-A3B-Instruct-2507-FP8 | Decoder Only | 31B | Apache-2.0 |
| Qwen/Qwen3-4B | Decoder Only | 4B | Apache-2.0 |
| Qwen/Qwen3-4B-Instruct-2507 | Decoder Only | 4B | Apache-2.0 |
| Qwen/Qwen3-4B-Thinking-2507 | Decoder Only | 4B | Apache-2.0 |
| Qwen/Qwen3-4B-Thinking-2507-FP8 | Decoder Only | 4B | Apache-2.0 |
| Qwen/Qwen3-8B | Decoder Only | 8B | Apache-2.0 |
| S4nfs/Neeto-1.0-8b | Decoder Only | 8B | CC-BY-NC-4.0 |
| SUSTech/SUS-Chat-34B | Decoder Only | 34B | Apache-2.0 |
| SciPhi/SciPhi-Mistral-7B-32k | Decoder Only | 7B | MIT |
| SciPhi/SciPhi-Self-RAG-Mistral-7B-32k | Decoder Only | 7B | MIT |
| Tesslate/UIGEN-X-4B-0729 | Decoder Only | 4B | Apache-2.0 |
| Tesslate/WEBGEN-4B-Preview | Decoder Only | 4B | Apache-2.0 |
| ThatSkyFox/DialoGPT-medium-whatsapp | Encoder Only | Unknown | Unknown |
| TheBloke/CodeLlama-70B-Instruct-AWQ | Decoder Only | 69B | Llama-2 |
| TheBloke/koala-13B-HF | Decoder Only | 13B | Other |
| TheBloke/tulu-30B-fp16 | Decoder Only | 30B | Other |
| TheMindExpansionNetwork/SYNERGETIC . . . | Decoder Only | 14B | Unknown |
| TheOneWhoWill/Bootstrap-LLM | Decoder Only | 0.1B | Apache-2.0 |
| Tianlin668/MentalBART | Encoder Decoder | Unknown | MIT |
| TigerResearch/tigerbot-13b-base | Decoder Only | 13B | Apache-2.0 |
| TinyLlama/TinyLlama-1.1B-Chat-v0.5 | Decoder Only | 1B | Apache-2.0 |
| TinyLlama/TinyLlama-1.1B-Chat-v1.0 | Decoder Only | 1B | Apache-2.0 |
| TinyLlama/TinyLlama-1.1B-intermediate-step- . . . | Decoder Only | 1B | Apache-2.0 |
| TinyLlama/TinyLlama_v1.1 | Decoder Only | 1B | Apache-2.0 |
| TinyLlama/TinyLlama_v1.1_math_code | Decoder Only | 1B | Apache-2.0 |
| Vikhrmodels/Vikhr-Nemo-12B-Instruct-R-21-09-24 | Decoder Only | 12B | Apache-2.0 |
| Vision-CAIR/vicuna | Decoder Only | Unknown | Unknown |
| Vortex5/Lunar-Nexus-12B | Decoder Only | 12B | Unknown |
| Vortex5/Moonlit-Shadow-12B | Decoder Only | 12B | Unknown |
| WhiteRabbitNeo/WhiteRabbitNeo-13B-v1 | Decoder Only | 13B | Llama-2 |
| WizardLM/WizardLM-70B-V1.0 | Decoder Only | 70B | Llama-2 |
| abocide/Qwen2.5-7B-Instruct-R1-forfinance | Decoder Only | 8B | Apache-2.0 |
| adamo1139/Apertus-8B-Instruct-2509-ungated | Decoder Only | 8B | Apache-2.0 |
| agentica-org/DeepCoder-14B-Preview | Decoder Only | 15B | MIT |
| allenai/OLMoE-1B-7B-0924 | Decoder Only | 7B | Apache-2.0 |
| allenai/tulu-2-dpo-70b | Decoder Only | 69B | Other |
| aquif-ai/aquif-3.5-8B-Think | Decoder Only | 8B | Apache-2.0 |
| askfjhaskjgh/UbermenschetienASI | Decoder Only | 8B | Other |
| augmxnt/shisa-base-7b-v1 | Decoder Only | 8B | Apache-2.0 |
| bardsai/jaskier-7b-dpo-v5.6 | Decoder Only | 7B | CC-BY-4.0 |
| berkeley-nest/Starling-LM-7B-alpha | Decoder Only | 7B | Apache-2.0 |
| beruniy/Llama-3.2-3B-Instruct-Uz | Decoder Only | 3B | llama3.2 |
| beyoru/Luna | Decoder Only | 4B | MIT |
| bigcode/octocoder | Decoder Only | 16B | OpenRAIL-M |
| bigcode/starcoder2-3b | Decoder Only | 3B | OpenRAIL-M |
| bigscience/bloom-560m | Decoder Only | 0.6B | BigScience RAIL 1.0 |
| bigscience/bloom-7b1 | Decoder Only | 7B | BigScience RAIL 1.0 |
| bigscience/bloomz-1b1 | Decoder Only | 1B | BigScience RAIL 1.0 |
| bigscience/bloomz-1b7 | Decoder Only | 2B | BigScience RAIL 1.0 |
| bigscience/bloomz-560m | Decoder Only | 0.6B | BigScience RAIL 1.0 |
| bigscience/bloomz-7b1 | Decoder Only | 7B | BigScience RAIL 1.0 |
| bxod/Llama-3.2-1B-Instruct-uz | Decoder Only | 1B | llama3.2 |
| chatitcloud/UZI1 | Decoder Only | 0.3B | Apache-2.0 |
| cisco-ai/mini-bart-g2p | Encoder Decoder | Unknown | Apache-2.0 |
| cloudyu/Mixtral_11Bx2_MoE_19B | Decoder Only | 19B | CC-BY-NC-4.0 |
| codefuse-ai/CodeFuse-DeepSeek-33B | Decoder Only | 33B | Other |
| codellama/CodeLlama-13b-Instruct-hf | Decoder Only | 13B | Llama-2 |

Table 8: Full list of used 305 models (continued)

| Model | Architecture | Parameters | License |
|---|---|---|---|
| codellama/CodeLlama-34b-Instruct-hf | Decoder Only | 34B | Llama-2 |
| codellama/CodeLlama-7b-hf | Decoder Only | 7B | Llama-2 |
| codeparrot/codeparrot-small-code-to-text | Encoder Only | Unknown | Apache-2.0 |
| codeparrot/codeparrot-small-multi | Encoder Only | Unknown | Apache-2.0 |
| continuedev/instinct | Decoder Only | 8B | Apache-2.0 |
| darkc0de/XortronCriminalComputingConfig | Decoder Only | 24B | Apache-2.0 |
| databricks/dolly-v1-6b | Decoder Only | 6B | CC-BY-NC-4.0 |
| databricks/dolly-v2-12b | Decoder Only | 12B | MIT |
| databricks/dolly-v2-3b | Decoder Only | 3B | MIT |
| databricks/dolly-v2-7b | Decoder Only | 7B | MIT |
| deepseek-ai/DeepSeek-Coder-V2-Lite-Instruct | Decoder Only | 16B | Other |
| deepseek-ai/DeepSeek-R1-0528-Qwen3-8B | Decoder Only | 8B | MIT |
| deepseek-ai/DeepSeek-R1-Distill-Llama-8B | Decoder Only | 8B | MIT |
| deepseek-ai/DeepSeek-R1-Distill-Qwen-1.5B | Decoder Only | 2B | MIT |
| deepseek-ai/DeepSeek-R1-Distill-Qwen-7B | Decoder Only | 8B | MIT |
| deepseek-ai/DeepSeek-V2-Lite | Decoder Only | 16B | Other |
| deepseek-ai/deepseek-coder-1.3b-base | Decoder Only | 1B | Other |
| deepseek-ai/deepseek-coder-6.7b-instruct | Decoder Only | 7B | Other |
| deepseek-ai/deepseek-llm-67b-chat | Decoder Only | 67B | Other |
| deepseek-ai/deepseek-llm-7b-base | Decoder Only | 7B | Other |
| deepseek-ai/deepseek-llm-7b-chat | Decoder Only | 7B | Other |
| deepseek-ai/deepseek-math-7b-instruct | Decoder Only | 7B | Other |
| dicta-il/dictalm2.0 | Decoder Only | 7B | Apache-2.0 |
| dphn/Dolphin-Mistral-24B-Venice-Edition | Decoder Only | 24B | Apache-2.0 |
| eci-io/climategpt-7b | Decoder Only | 7B | Other |
| elyza/ELYZA-japanese-Llama-2-7b | Decoder Only | 7B | Llama-2 |
| elyza/ELYZA-japanese-Llama-2-7b-instruct | Decoder Only | 7B | Llama-2 |
| elyza/Llama-3-ELYZA-JP-8B | Decoder Only | 8B | llama3 |
| eren23/ogno-monarch-jaskier-merge-7b-OH-PREF-… | Decoder Only | 7B | CC-BY-NC-4.0 |
| facebook/galactica-1.3b | Decoder Only | 1B | CC-BY-NC-4.0 |
| fblgit/UNA-SimpleSmaug-34b-v1beta | Decoder Only | 34B | Apache-2.0 |
| fluently/FluentlyQwen3-1.7B | Decoder Only | 2B | Apache-2.0 |
| futurehouse/ether0 | Decoder Only | 24B | Apache-2.0 |
| golaxy/gowizardlm | Decoder Only | Unknown | Apache-2.0 |
| google-t5/t5-large | Encoder Decoder | 0.7B | Apache-2.0 |
| google-t5/t5-small | Encoder Decoder | 60.5M | Apache-2.0 |
| google/flan-t5-base | Encoder Decoder | 0.2B | Apache-2.0 |
| google/flan-t5-large | Encoder Decoder | 0.8B | Apache-2.0 |
| google/gemma-2b-it | Unknown | 3B | Gemma |
| google/gemma-3-12b-it | Unknown | 12B | Gemma |
| google/gemma-3-1b-it | Unknown | 1B | Gemma |
| google/gemma-3-4b-it | Unknown | 4B | Gemma |
| google/gemma-7b-it | Unknown | 9B | Gemma |
| gradientai/Llama-3-8B-Instruct-262k | Decoder Only | 8B | llama3 |
| heegyu/kogpt-j-base | Decoder Only | Unknown | MIT |
| huggyllama/llama-7b | Decoder Only | 7B | Other |
| huihui-ai/Huihui-Jan-v1-4B-abliterated | Decoder Only | 4B | Apache-2.0 |
| ibivibiv/alpaca-dragon-72b-v1 | Decoder Only | 72B | Other |
| inflatebot/MN-12B-Mag-Mell-R1 | Decoder Only | 12B | Unknown |
| janhq/Jan-v1-4B | Decoder Only | 4B | Apache-2.0 |
| janhq/Jan-v1-edge | Decoder Only | 2B | Apache-2.0 |
| jiawei-ucas/Qwen-2.5-7B-ConsistentChat | Decoder Only | 8B | MIT |
| jxm/gpt-oss-20b-base | Unknown | 20B | Unknown |
| kevin009/llamaRAGdrama | Decoder Only | 7B | Apache-2.0 |
| knifeayumu/Cydonia-v4.1-MS3.2-Magnum-… | Decoder Only | 24B | Apache-2.0 |
| kyujinpy/Sakura-SOLRCA-Math-Instruct-DPO-v1 | Decoder Only | 11B | CC-BY-NC-SA-4.0 |
| lgaalves/gpt1 | Decoder Only | 0.1B | MIT |
| liyuesen/druggpt | Encoder Only | Unknown | GPL-3.0 |
| lmsys/vicuna-13b-v1.5 | Decoder Only | 13B | Llama-2 |
| lmsys/vicuna-33b-v1.3 | Decoder Only | 33B | Unknown |
| lmsys/vicuna-7b-v1.1 | Decoder Only | 7B | Unknown |
| lmsys/vicuna-7b-v1.5 | Decoder Only | 7B | Llama-2 |
| lmsys/vicuna-7b-v1.5-16k | Decoder Only | 7B | Llama-2 |
| m-a-p/ChatMusician | Decoder Only | 7B | MIT |
| meta-llama/CodeLlama-13b-Instruct-hf | Unknown | 13B | Llama-2 |
| meta-llama/CodeLlama-7b-Instruct-hf | Unknown | 7B | Llama-2 |
| meta-llama/Llama-2-13b-chat-hf | Unknown | 13B | Llama-2 |
| meta-llama/Llama-2-70b-chat-hf | Unknown | 69B | Llama-2 |
| meta-llama/Llama-2-7b-chat-hf | Unknown | 7B | Llama-2 |
| meta-llama/Llama-2-7b-hf | Unknown | 7B | Llama-2 |
| meta-llama/Llama-3.1-8B-Instruct | Unknown | 8B | llama3.1 |
| meta-llama/Llama-3.2-1B-Instruct | Unknown | 1B | llama3.2 |
| meta-llama/Llama-3.2-3B-Instruct | Unknown | 3B | llama3.2 |
| meta-llama/LlamaGuard-7b | Unknown | 7B | Llama-2 |
| meta-llama/Meta-Llama-3-70B | Unknown | 71B | llama3 |
| meta-llama/Meta-Llama-3-70B-Instruct | Unknown | 71B | llama3 |
| meta-llama/Meta-Llama-3-8B | Unknown | 8B | llama3 |
| meta-llama/Meta-Llama-3-8B-Instruct | Unknown | 8B | llama3 |

Table 8: Full list of used 305 models (continued)

| Model | Architecture | Parameters | License |
|---|---|---|---|
| meta-llama/Meta-Llama-Guard-2-8B | Unknown | 8B | llama3 |
| meta-math/MetaMath-Llemma-7B | Decoder Only | 7B | Apache-2.0 |
| meta-math/MetaMath-Mistral-7B | Decoder Only | 7B | Apache-2.0 |
| microsoft/DialoGPT-medium | Encoder Only | Unknown | MIT |
| microsoft/MediPhi-Instruct | Decoder Only | 4B | MIT |
| microsoft/Orca-2-13b | Decoder Only | 13B | Other |
| microsoft/Orca-2-7b | Decoder Only | 7B | Other |
| microsoft/Phi-3-mini-128k-instruct | Decoder Only | 4B | MIT |
| microsoft/Phi-3-mini-4k-instruct | Decoder Only | 4B | MIT |
| microsoft/Phi-3.5-mini-instruct | Decoder Only | 4B | MIT |
| microsoft/Phi-4-mini-instruct | Decoder Only | 4B | MIT |
| microsoft/Phi-4-mini-reasoning | Decoder Only | 4B | MIT |
| microsoft/bitnet-b1.58-2B-4T | Decoder Only | 0.8B | MIT |
| microsoft/phi-1 | Decoder Only | 1B | MIT |
| microsoft/phi-1_5 | Decoder Only | 1B | MIT |
| microsoft/phi-2 | Decoder Only | 3B | MIT |
| microsoft/phi-4 | Decoder Only | 15B | MIT |
| miromind-ai/MiroThinker-14B-DPO-v0.2 | Decoder Only | 14B | Apache-2.0 |
| mistralai/Ministral-8B-Instruct-2410 | Unknown | 8B | Other |
| mistralai/Mistral-7B-Instruct-v0.1 | Unknown | 7B | Apache-2.0 |
| mistralai/Mistral-7B-Instruct-v0.3 | Unknown | 7B | Apache-2.0 |
| mistralai/Mixtral-8x7B-Instruct-v0.1 | Unknown | 47B | Apache-2.0 |
| mlabonne/AlphaMonarch-7B | Decoder Only | 7B | CC-BY-NC-4.0 |
| mlx-community/Apertus-8B-Instruct-2509-bf16 | Decoder Only | 8B | Apache-2.0 |
| mookiezi/Discord-Micae-8B-Preview | Decoder Only | 8B | llama3 |
| mookiezi/Discord-Micae-Hermes-3-3B | Decoder Only | 3B | llama3 |
| mosaicml/mpt-30b-chat | Decoder Only | 30B | CC-BY-NC-SA-4.0 |
| mosaicml/mpt-30b-instruct | Decoder Only | 30B | Apache-2.0 |
| mosaicml/mpt-7b-storywriter | Decoder Only | 7B | Apache-2.0 |
| nakodanei/Blue-Orchid-2x7b | Decoder Only | 13B | Apache-2.0 |
| nomic-ai/gpt4all-13b-snoozy | Decoder Only | 13B | gpl |
| nvidia/OpenReasoning-Nemotron-1.5B | Decoder Only | 2B | CC-BY-4.0 |
| openai-community/openai-gpt | Decoder Only | 0.1B | MIT |
| openai/gpt-oss-20b | Decoder Only | 22B | Apache-2.0 |
| openbmb/UltraCM-13b | Decoder Only | 13B | MIT |
| openchat/openchat-3.5-0106 | Decoder Only | 7B | Apache-2.0 |
| openchat/openchat_3.5 | Decoder Only | 7B | Apache-2.0 |
| openlm-research/open_llama_13b | Decoder Only | 13B | Apache-2.0 |
| openlm-research/open_llama_7b | Decoder Only | 7B | Apache-2.0 |
| prithivMLmods/rStar-Coder-Qwen3-0.6B | Decoder Only | 0.6B | Apache-2.0 |
| project-baize/baize-v2-13b | Decoder Only | 13B | CC-BY-NC-4.0 |
| quantumaikr/KoreanLM | Decoder Only | 7B | Unknown |
| rishiraj/CatPPT-base | Decoder Only | 7B | Apache-2.0 |
| roneneldan/TinyStories-1M | Decoder Only | 1M | Unknown |
| sail/Sailor-7B | Decoder Only | 8B | Apache-2.0 |
| sarvamai/sarvam-1 | Decoder Only | 3B | Unknown |
| scb10x/typhoon-7b | Decoder Only | 7B | Apache-2.0 |
| segolilylabs/Lily-Cybersecurity-7B-v0.2 | Decoder Only | 7B | Apache-2.0 |
| sequelbox/gpt-oss-20b-DES-Reasoning | Decoder Only | 21B | Apache-2.0 |
| stabilityai/stablelm-tuned-alpha-7b | Decoder Only | 7B | CC-BY-NC-SA-4.0 |
| stanford-crfm/BioMedLM | Encoder Only | Unknown | BigScience RAIL 1.0 |
| tencent/Hunyuan-1.8B-Instruct | Decoder Only | 2B | Unknown |
| tencent/Hunyuan-7B-Instruct | Decoder Only | 8B | Unknown |
| theprint/TiTan-Qwen2.5-0.5B | Decoder Only | 0.5B | Apache-2.0 |
| thesephist/contra-bottleneck-t5-large-wikipedia | Encoder Decoder | Unknown | MIT |
| tiiuae/Falcon3-10B-Instruct | Decoder Only | 10B | Other |
| tiiuae/Falcon3-1B-Instruct | Decoder Only | 2B | Other |
| tiiuae/Falcon3-3B-Instruct | Decoder Only | 3B | Other |
| tiiuae/Falcon3-7B-Instruct | Decoder Only | 7B | Other |
| tiiuae/falcon-11B | Decoder Only | 11B | Unknown |
| tiiuae/falcon-40b-instruct | Decoder Only | 40B | Apache-2.0 |
| tiiuae/falcon-7b-instruct | Decoder Only | 7B | Apache-2.0 |
| tomg-group-umd/DynaGuard-1.7B | Decoder Only | 2B | Apache-2.0 |
| tomg-group-umd/DynaGuard-4B | Decoder Only | 4B | Apache-2.0 |
| tomg-group-umd/DynaGuard-8B | Decoder Only | 8B | Apache-2.0 |
| unsloth/gpt-oss-20b-BF16 | Decoder Only | 21B | Apache-2.0 |
| upstage/SOLAR-10.7B-Instruct-v1.0 | Decoder Only | 11B | CC-BY-NC-4.0 |
| upstage/SOLAR-10.7B-v1.0 | Decoder Only | 11B | Apache-2.0 |
| vilm/vinallama-7b-chat | Decoder Only | 7B | Llama-2 |
| wave-on-discord/silly-v0.2 | Decoder Only | 12B | Apache-2.0 |
| yam-peleg/Experiment26-7B | Decoder Only | 7B | Apache-2.0 |
| zai-org/GLM-4-32B-0414 | Decoder Only | 33B | MIT |
| zai-org/glm-4-9b-chat-hf | Decoder Only | 9B | Other |
| zhengr/MixTAO-7Bx2-MoE-v8.1 | Decoder Only | 13B | Apache-2.0 |

