# OpenReview forum: "LLM DNA: Tracing Model Evolution via Functional Representations"
_ICLR.cc/2026/Conference — ICLR 2026 Oral_

### Official Review · Reviewer_ZryQ · 2025-10-22

**Soundness:** 3
**Presentation:** 3
**Contribution:** 2
**Rating:** 6
**Confidence:** 3

**Summary:**

This paper defines a DNA to LLMs in order to build a distance metric between LLMs to produce very interesting mapping at a fairly low cost (only need a forward pass and no need to retrain a model when adding a new LLM to the map).

**Strengths:**

- importing concepts from genetics to apply them to LLMs is inspiring
- the method is well documented and math supports the methods
- the subject addressed is important as more and more models are published everyday with lower and lower transparency about training details especially for private models

**Weaknesses:**

- **Important related work missing** : This paper adresses a nearly identical topic with similar methods than another paper published last year in the same conference https://iclr.cc/virtual/2025/poster/28195 without discussing and comparing with it. An in depth comparison of both methods would be very important as it would provide a good baseline to compare both the theoretical framework as well as results in practice increasing the impact of the paper.
- **Definition of LLM evolution unclear** :The paper defined clearly what is genetic determinism and inheritance but didn't clearly define what is evolution in LLMs while this is a crucial concept in the paper. Is it a model A finetuned from model B ? Is it the general evolution of pretraining sets ? Is it something else ? This concept is central to the analogy and could deserve a much more in depth explanation strengthening even further the already strong basis of the paper.
- **No limitations discussion** : are there cases where the algorithm doesn't work as expected ? It is important to discuss the upsides of the methods but also the parts where it doesn't perform particularly well while explaining why. This can only make the paper more interesting for the reader as well as fostering follow up work.

**Questions:**

- The method presented used classical benchmarks to test the completion of models on these benchmarks. Some models are suspected to be contaminated on classical benchmarks like MMLU. This could bias the metric making models with different origins appear with a similar DNA. Wouldn't it be better to use text not included in classical benchmarks to avoid this ?
- The algorithm seems to capture very well some relationships between models in Figure 4. However some relationships are not captured such as Meta-Llama-3-8B that is not close to Meta-Llama-3-8B-Instruct while the latter is likely finetuned from the former. What could cause this difference ? (there are many other examples of models that are not close to each other in the figure)
- In Figure 4, clusters are based on organizations that trained the models. However similarity between LLMs could either come from a similar base model (but trained on different finetuning sets) or because they come from different pretraining sets (but were finetuned on the same finetuning set). How does this plays out in the results ? Does one scenario lead to much higher similarity than the other ?

---

> ### Author Response · Authors · 2025-11-23
> **Author Response (Part 1)**
>
> We thank Reviewer `ZryQ` for your insightful and constructive comments, which have greatly helped improve this paper. We address all concerns below, and the corresponding revisions are included in the **revised paper** (see the updated PDF with **changes highlighted in blue**). For convenience, we also paste the revised text (excluding figures and tables) in our response.
>
>
> ---
>
> ### Weakness 1: "Missing related work (PhyloLM, ICLR 2025) and comparison".
>
> In the revision, we include PhyloLM as a baseline in **Tables 1 and 4**. The results show that LLM DNA **significantly and consistently outperforms PhyloLM** in relation prediction. This is expected because PhyloLM computes distances from token distributions, and its proposed workaround does not generalize well to multi-tokenizer LLMs. We also add a comparison to PhyloLM in the **related-work section**.
>
>
> > **Section 5.2: Detect LLM Relationships**
> >
> >**Table 1: DNA LLM-relation detection test performance (mean ±standard deviation across five seeds)**
> >|Method|Acc|Prec|Rec|F1|AUC|
> >|:-|-:|-:|-:|-:|-:|
> >|Random |0.493±0.036 |0.515±0.034|0.516±0.042|0.516±0.036|0.492±0.036|
> >|Greedy |0.593±0.018 |0.759±0.081|0.329±0.000|0.458±0.014|0.606±0.023|
> >|PhyloLM + SVM |0.742±0.058 |0.741±0.073 |0.812±0.123|0.766±0.057|0.788±0.060|
> >|DNA (Qwen3) + SVM |0.919±0.048 |0.898±0.049 |0.957±0.072 |0.925±0.047|0.979±0.027|
> >|DNA (BGE) + SVM|**0.934±0.053** |0.905±0.068|**0.982±0.059**|**0.940±0.050** |**0.992±0.023** |
> > | DNA (MPNet) + SVM|0.932±0.048|**0.914±0.065**|0.966±0.069|0.937±0.047|0.990±0.020|
> >
> > **Section 2: Related Work**
> > Yax et al. (2025) propose PhyloLM, which constructs a phylogenetic tree using a token-distribution-based distance between LLMs. However, this distance is token-based and does not generalize well to heterogeneous LLMs with different tokenizers.
>
> ---
>
> ### Weakness 2: "Definition of LLM evolution unclear"
>
> We define evolution as a bounded function on LLMs, with a formal mathematical definition provided in **Appendix A.1 of the revised paper**.
>
> > Before discussing the properties, we begin by defining the evolution of an LLM.
> >
> > **Definition A.1 (Evolution).**
> > Let $\mathcal{F}$ be the LLM function space in Definition 1, and let $d_H$ be the Hilbert distance on $\mathcal{F}$. For a threshold $\delta_H>0$, an **evolution** is a function $\mathcal{E}:\mathcal{F}\rightarrow \mathcal{F}$ such that for any $f\in\mathcal{F}$,
> >
> > $$d_H\big(f,\mathcal{E}(f)\big)<\delta_H.$$
> >
> > If $f_2=\mathcal{E}(f_1)$, we say $f_2$ is evolved from $f_1$ within $\delta_H$.
> >
> > This notion covers a wide range of operations that induce a bounded shift in an LLM’s functional behavior, including but not limited to fine-tuning, distillation, and reinforcement learning, as they modify the LLM within a bounded Hilbert-distance threshold.
>
> ---
>
> ### Weakness 3. "Missing limitations discussion."
>
> In the revised paper, we include **Appendix E** for limitations.
>
> > **Appendix E: Limitaiton**
> >
> > This section discusses the limitations of our work.
> >
> > **Traits and DNA subsequences.** Our formal definition of LLM DNA does not assign mathematical meaning to DNA subsequences. However, in Section 5.6, we observe that fine-tuning alters only specific subsequences while leaving the overall pattern largely intact. This suggests that **certain DNA subsequences may encode distinct model traits** (e.g., mathematical reasoning). A theoretical characterization of how different evolutions affect such traits is an important and challenging direction for future work. In the long run, this could enable more precise understanding, management, and even targeted modification of LLM DNA, drawing a closer parallel to how biogenetics studies and manipulates inherited properties.
> >
> > **Adaptive attacks on DNA.** Our current DNA extraction is not designed to be robust against adaptive attacks. If an attacker knows the extraction data, they could deliberately train a model toward a malicious objective—e.g., producing a substantially different DNA while functionally copying another model—in order to evade lineage detection, patents, or license constraints. A straightforward mitigation is to keep extraction datasets closed-source or to re-extract DNA using a fresh dataset when needed. A more principled direction is to scale the extraction set so that overfitting to it becomes prohibitively expensive and would noticeably degrade general performance. Developing such security guarantees remains a key topic for future work.
> >

---

> ### Author Response · Authors · 2025-11-23
> **Author Response (Part 2)**
>
> > **Data bias and contamination.** Our DNA extraction relies on six public datasets, which may introduce evaluation bias or contamination. Bias could cause us to overlook certain differences or overestimate similarity among some model groups, while contamination could arise if parts of the extraction data were seen during pretraining. **Nonetheless, their impact on our evaluation is limited**. As shown in Appendix C.7, DNA remains effective even when evaluated on purely random input data. This observation supports that DNA primarily captures *relative relationships* among LLMs, rather than depending on absolute accuracy on any particular dataset.
>
> ---
>
> ### Question 1: "Benchmark contamination (MMLU) might bias the metric. Better to use non-benchmark text?"
>
> Data contamination does not substantially affect our evaluation because LLM DNA targets **relative relationships** among LLMs rather than absolute performance. For instance, if Models A and B are both contaminated on MMLU, they will each produce correct answers with high probability; DNA will still group them together as expected.
>
> Moreover, **DNA is insensitive to the evaluated datasets**. We confirm this by an additional experiment in the **Appendix C.7 of revised paper**. The results show that even replacing the standard benchmark inputs with randomly generated data does not degrade the quality of DNA. This is because DNA analyses (e.g., relationship prediction) target *relative relationships* among LLMs rather than absolute accuracy.
>
>
> >**Appendix C.7: DNA Extracted from Random Data**
> > This subsection demonstrates the strong robustness of LLM DNA: it preserves distinguishability even when the evaluation inputs are random. Specifically, we use `wonderwords.RandomWord` to generate 600 samples (matching our main evaluation: 6 datasets with 100 samples each). Each sample is a 100-word sentence. We run inference with each LLM on this dataset, collect their outputs, and encode them using Qwen3-8B-Embedding, identical to the main DNA extraction process. We then perform the relation prediction task using these DNAs. An example slice of the random input data is:
> >
> > ```text
> > disdain chapel intention gymnast activation nymph prickly chalk sad method historical ferryboat majority hybridization ... parachute ferryboat download aim scout cutting x-ray shack codpiece hypothesis endothelium masonry
> > ```
> >
> > The results are reported in Table 4. Interestingly, the **random-data setting even slightly outperforms the standard benchmark**. This highlights that the DNA extraction method is largely insensitive to input-dataset bias. The key intuition is that DNA analyses (e.g., relationship prediction) target *relative relationships* among LLMs rather than absolute accuracy. Different datasets act as different “views” of the same model set: a suboptimal view may obscure some relations, but it is unlikely to fabricate spurious ones—for instance, it will not make two genuinely similar models appear unrelated. This observation also aligns with the stability we see under random projection.
> >
> > **Table 4: DNA (Qwen3) relation prediction performance (DNA extracted from random inputs)**
> >
> > | Method | Accuracy | Precision | Recall | F1 | AUC|
> > | - | - | - | - | - | - |
> > | PhyloLM + SVM| 0.742 ± 0.058|0.741 ± 0.073|0.812 ± 0.123|0.766 ± 0.057|0.788 ± 0.060|
> > | DNA + SVM| *0.919 ± 0.048*   |*0.898 ± 0.049*|*0.957 ± 0.072*|*0.925 ± 0.047*|*0.979 ± 0.027*|
> > | DNA (Rand) + SVM |**0.949 ± 0.048**|**0.931 ± 0.053**|**0.977 ± 0.062**|**0.952 ± 0.048** | **0.987 ± 0.038** |
>
>
> ---
>
>
> ### Question 2: "Why are Meta-Llama-3-8B and Llama-3-8B-Instruct not close in Figure 4"?
>
> Because **Llama-3-8B-Instruct is instruction-tuned**, whereas **Meta-Llama-3-8B is a base text-completion model**. Instruction tuning changes the model’s functional behavior (and thus its DNA) substantially: it shifts the output distribution from free-form continuation to dialogue-style answering. Concretely:
>
> * **Instruction-tuned model (conversation)**
>   Input: “The capital of France” → Output: “The capital of France is Paris. Would you like to know more?”
> * **Text-completion model (continuation)**
>   Input: “The capital of France” → Output: “is Paris and it is beautiful …”
>
> These outputs correspond to different patterns, so their functional embeddings are far apart even though they share the same pretraining backbone. The same split is visible in Table 7, where base text-completion models cluster on one major branch and instruction-tuned variants on another.
>
> ---

---

> ### Author Response · Authors · 2025-11-23
> **Author Response (Part 3)**
>
> ### Question 3. Does similarity come from Base Model (Pre-training) or Fine-tuning data?
>
> Whether pretraining or fine-tuning dominates similarity depends primarily on **the amount and distribution of the data** used in each stage. Our results reflect **both** scenarios:
>
> 1. **Base-model lineage usually dominates.**
>    In most modern LLMs, fine-tuning datasets are orders of magnitude smaller than pretraining corpora. As a result, fine-tuned variants tend to remain **near their base models** in DNA space in Figure 4, inheriting most functional traits from pretraining. This is why models from the same base family often cluster tightly even when their instruction/adapter sets differ.
> 2. **Fine-tuning can dominate when it introduces a strong distribution shift.**
>    When fine-tuning data has a **distinct format or objective** (e.g., instruction tuning), it can substantially reshape functional behaviors and push models **far from the base branch**. This is exactly what we observe for instruction-tuned vs. text-completion models: despite sharing a base, they separate into different major branches.

---

> ### Comment · Reviewer_ZryQ · 2025-11-23
> **Reviewer Response**
>
> I thank the authors for their thorough and comprehensive response. However, I still have a few remaining concerns:
>
> **Regarding Weakness 1**: I appreciate the discussion of the comparison with PhyloLM. However, the comments made by the authors about the fact that PhyloLM doesn't work well on multi-tokeniser LLMs make me think they may not have implemented the algorithm properly. Indeed, PhyloLM specifically used only 4-characters long completions as a universal tokenizer instead of individual LLM tokenizers to not face this multi tokeniser comparison limitation (see section 2.4). Therefore, the results presented may not constitute a fully accurate baseline comparison with PhyloLM since they seem to implement a different algorithm. Fixing this implementation would highlight more accurately the advantages/drawbacks of the authors algorithm with respect to current state of the art methods.
>
> **Regarding Question 2**: Chat-tuned models often exhibit distinctive behaviors like the one cited when queried with chat-specific formatting tokens (e.g., `<|im_start|>[role]`). I am curious whether the authors have experimented with querying these models using a completion format instead of a chat format (without these chat tokens). This alternative approach might enhance the effectiveness of the algorithm and could be worth exploring.

---

> > ### Author Response · Authors · 2025-11-24
> > **Response to Follow-up (Part 1)**
> >
> > We thank Reviewer `ZryQ` for reading our response carefully and for raising insightful follow-up questions. We address both questions below, and the corresponding changes have been integrated into **Appendix C.8 and C.9** of the revised paper.
> >
> >
> > ### **Follow-up 1: Compare with PhyloLM on LLM set with the same tokenizer**
> >
> > We acknowledge your point regarding PhyloLM's "proxy" method (using 4-character completions) to handle tokenizer discrepancies. To address this concern rigorously, rather than relying on the proxy approximation, **we conducted a new experiment in the "ideal" setting for PhyloLM: a subset of LLMs that share the exact same tokenizer** (Qwen2Tokenizer). This setting eliminates the need for the 4-character proxy and allows PhyloLM to operate on exact token distributions (its native input). As detailed in the **new Appendix C.8**, while PhyloLM’s performance indeed improves in this single-tokenizer setting (accuracy 0.742 → 0.755, precision 0.74 → 0.83), **LLM DNA still significantly outperforms it (accuracy 0.834 for DNA-Qwen3, 0.869 for DNA-BGE).** This confirms that even when PhyloLM is given its optimal operating conditions, LLM DNA provides a more distinct functional signature.
> >
> > > **Appendix C.8: Relation Prediction for LLMs with Same Tokenizer**
> > >
> > >Since PhyloLM (Yax et al., 2025) is designed to compare distances between LLMs that share the same tokenizer, we further conduct an experiment that compares PhyloLM with LLM DNA on a same-tokenizer subset. This setting is more challenging because models with the same tokenizer are typically closer in function and thus harder to distinguish. Concretely, we filter the dataset to include only models using Qwen2Tokenizer, the most commonly adopted tokenizer among our evaluated LLMs, and perform the same relation prediction task. The results are reported in Table 6.
> > >
> > >Comparing Table 6 with Table 1 yields several key findings. First, **all methods except PhyloLM exhibit a clear performance drop in the same-tokenizer setting**: for example, DNA (MPNet) decreases from 0.93 to 0.86 in accuracy, and Greedy drops from 0.59 to 0.42. In contrast, PhyloLM slightly improves its accuracy (0.742 to 0.755) and achieves a notable precision gain (0.74 to 0.83), confirming that the same-tokenizer case is indeed more challenging while PhyloLM remains robust. Second, **despite the reduced gap, PhyloLM is still consistently outperformed by DNA**. This may be because PhyloLM primarily evaluates the distribution of the first token, which may be insufficient to capture long-range sentence-level distributions that are better reflected in DNA.
> > >
> > >
> > > **Table 6: DNA relation prediction performance for LLMs with Qwen2Tokenizer**
> > >
> > > | Method                |          Accuracy |         Precision |            Recall |                F1 |               AUC |
> > > | --------------------- | ----------------: | ----------------: | ----------------: | ----------------: | ----------------: |
> > > | Random                |     0.511 ± 0.054 |     0.682 ± 0.073 |     0.529 ± 0.054 |     0.593 ± 0.047 |     0.501 ± 0.068 |
> > > | Greedy                |     0.428 ± 0.043 |     0.687 ± 0.101 |     0.294 ± 0.000 |     0.410 ± 0.018 |     0.500 ± 0.049 |
> > > | PhyloLM + SVM         |     0.755 ± 0.136 |     0.832 ± 0.080 |     0.800 ± 0.204 |     0.801 ± 0.149 |     0.760 ± 0.180 |
> > > | **DNA (Qwen3) + SVM** |     0.834 ± 0.092 |     0.863 ± 0.073 |     0.907 ± 0.123 |     0.878 ± 0.072 |     0.926 ± 0.105 |
> > > | **DNA (BGE) + SVM**   | **0.869 ± 0.078** |   *0.865 ± 0.069* | **0.966 ± 0.094** | **0.909 ± 0.059** |   *0.944 ± 0.087* |
> > > | **DNA (MPNet) + SVM** |   *0.866 ± 0.088* | **0.879 ± 0.099** |   *0.948 ± 0.098* |   *0.906 ± 0.065* | **0.960 ± 0.079** |
> >
> > ---

---

> ### Author Response · Authors · 2025-11-24
> **Response to Follow-up (Part 2)**
>
> ### **Follow-up 2: Chat-tuned model with completion format**
>
> We are sincerely grateful for this thoughtful and constructive suggestion. It directly improved both the fairness and clarity of our evaluation. Following your idea, we remove chat templates for instruction-tuned LLMs during DNA extraction and rerun the experiments in Table 1. The updated results show that **removing the chat template consistently improves accuracy by 1–3% across encoders**. This is exactly because evaluating chat-tuned models in a completion-style interface yields a fairer and more robust comparison with their base models. We will adopt this setting as the **default in our DNA extraction pipeline**. The new ablation study is reported in **Table 7 and Appendix C.9** of the revised paper. We truly appreciate your contribution to strengthening our methodology and will explicitly acknowledge your input in the final version of the paper.
>
> > **Appendix C.9: Ablation Study for Chat Templates**
> >
> > This subsection studies the effect of chat templates on instruction-tuned models. A chat template typically inserts special tokens and formatting used during instruction tuning to teach the model conversational behavior. For an instruction-tuned model (e.g., `Llama3-Instruct`), we consider two settings: (i) feeding the raw prompt directly to the model without applying the template, which makes it behave like a completion model; and (ii) applying the template and extracting only the model’s response for DNA computation. For non-chat (base/completion) models, the DNA extraction procedure is unchanged and consistent.
> >
> > We report relationship prediction performance (under the same setting as Table 1) in Table 7. **Removing the chat template consistently improves accuracy by 1--3% across encoders.** This indicates that, for cross-family comparisons involving both chat-tuned and completion models, omitting templates yields a fairer and more direct functional comparison, since chat-tuned models are evaluated in the same completion-style interface as their base LLMs.
> >
> > **Table 7: DNA relation prediction performance: with vs without chat templates**
> >
> > | Encoder | Template     | Accuracy          | Precision         | Recall            | F1                | AUC               |
> > | ------- | ------------ | ----------------- | ----------------- | ----------------- | ----------------- | ----------------- |
> > | Qwen3   | w/ template  | 0.919 ± 0.048     | 0.898 ± 0.049     | 0.957 ± 0.072     | 0.925 ± 0.047     | 0.979 ± 0.027     |
> > | Qwen3   | w/o template | **0.930 ± 0.045** | **0.916 ± 0.067** | **0.961 ± 0.071** | **0.935 ± 0.044** | **0.994 ± 0.021** |
> > | MPNet   | w/ template  | 0.932 ± 0.048     | 0.914 ± 0.065     | 0.966 ± 0.069     | 0.937 ± 0.047     | 0.990 ± 0.020     |
> > | MPNet   | w/o template | **0.958 ± 0.047** | **0.943 ± 0.060** | **0.982 ± 0.059** | **0.960 ± 0.047** | **0.994 ± 0.016** |
> > | BGE     | w/ template  | 0.934 ± 0.053     | 0.905 ± 0.068     | **0.982 ± 0.059** | 0.940 ± 0.050     | **0.992 ± 0.023** |
> > | BGE     | w/o template | **0.945 ± 0.054** | **0.929 ± 0.073** | 0.974 ± 0.066     | **0.948 ± 0.052** | **0.992 ± 0.023** |
>
> ---

---

> > ### Comment · Reviewer_ZryQ · 2025-11-24
> > **Follow Up**
> >
> > I thank the authors for their answer. I think this is indeed a more accurate way to compare the two algorithms. One last note is that technically PhyloLM was designed to address LLMs with different tokenizers (with this 2.4 section) so I believe this statement
> > >Since PhyloLM (Yax et al., 2025) is designed to compare distances between LLMs that share the same tokenizer
> >
> > is not perfectly accurate and a better phrasing could be found to introduce the comparison method and its limitations. Indeed, PhyloLM was designed to take into account multiple tokenizer and the comparison here used the model's tokenizer rather than the first 4 characters which is probably similar but may not lead to the exact same results. This first comparison between the two methods is still really good and highlights the interest for DNA but I believe it is very important for transparency, reproducibility and scientific clarity to clearly acknowledge both
> > - the fact that PhyloLM was designed to take into account multiple tokenizers
> > - and this comparison used Qwen's tokenizer to align the two methods instead of using the first 4 characters which is technically a slightly different version of PhyloLM
> >
> > Even if the comparison is not perfect, the authors may say that future work could better explore the comparison (maybe also using the text datasets used to design PhyloLM rather than the one designed for DNA for example, ...). In addition to this revised Appendix, I hope the authors have also updated the related work section.
> >
> > On the other hand, I found this part very interesting
> > >This may be because PhyloLM primarily evaluates the distribution of the first token, which may be insufficient to capture long-range sentence-level distributions that are better reflected in DNA.
> >
> > but its in-depth investigation should probably be kept for future work.
> >
> > Other than these remarks about comparison to PhyloLM (the fact that it was technically designed for multiple tokenizers + acknowledging the limitations of the comparison) the authors very accurately addressed most of my concerns and I am ready to increase my grade once these small framing details about transparency of the method and results are corrected.

---

> ### Author Response · Authors · 2025-11-24
> **Author Response to Follow-up**
>
> We appreciate the point you raised and fully agree that transparency, reproducibility, and scientific clarity are crucial. Following your advice, we have revised **both the related work and Appendix C.8** to correct the inaccuracies you identified.
>
> The revised text now clearly acknowledges that PhyloLM computes general LLM distances while accounting for multiple tokenizers, and it explicitly states the limitations of our current evaluation. We also outline future directions for a fairer and more comprehensive comparison—e.g., evaluating on PhyloLM’s original datasets and tasks—as well as for investigating how sentence-level, long-range patterns influence LLM representations.
>
> All these changes have been incorporated into the **revised paper (see attached PDF)**, and the modified contents are pasted below for convenience (with changes in **bold**).
>
>
> >### **Section 2: Related Work**
> >
> >...
> >**Yax et al. (2025) propose PhyloLM, a state-of-the-art method for constructing phylogenetic trees of LLMs. PhyloLM estimates inter-model distances via a coupled design tailored to tokenizer compatibility: for LLMs sharing the same tokenizer, it computes Nei’s genetic distance between their output token distributions; for LLMs with different tokenizers, it instead relies on “the first 4 characters in the concatenation of these tokens.” This distance largely reflects first-token behavior, which may be limited in its ability to capture long-range, sentence-level distributions.** In contrast, our LLM DNA is a stable and generalizable sentenced-level intrinsic property extractable independently for each LLM.
>
>
> >### **Appendix C.8: Relation Prediction for LLMs with Same Tokenizer**
> >
> >**Though our state-of-the-art baseline PhyloLM (Yax et al., 2025) is designed to compare general LLMs across both same- and different-tokenizer settings, it introduces different comparison strategies for these two cases. In particular, when models share a tokenizer, PhyloLM can directly leverage exact token distributions, which typically yields stronger performance than its cross-tokenizer variant based on the first four characters (as shown in Table 1). To provide a fair and transparent comparison under PhyloLM’s same-tokenizer regime, we further compare PhyloLM with LLM DNA on a single-tokenizer subset.** This setting is more challenging because models with the same tokenizer are typically closer in function and thus harder to distinguish. Concretely, we filter the dataset to include only models using Qwen2Tokenizer, the most commonly adopted tokenizer among our evaluated LLMs, and perform the same relation prediction task. The results are reported in Table 6.
> >
> >Comparing Table 6 with Table 1 yields several key findings. First, all methods except PhyloLM exhibit a clear performance drop in the same-tokenizer setting: for example, DNA (MPNet) decreases from 0.93 to 0.86 in accuracy, and Greedy drops from 0.59 to 0.42. In contrast, PhyloLM slightly improves its accuracy (0.742 to 0.755) and achieves a notable precision gain (0.74 to 0.83), confirming that the same-tokenizer case is indeed more challenging while PhyloLM remains robust. Second, despite the reduced gap, PhyloLM is still consistently outperformed by DNA. This may be because PhyloLM primarily evaluates the distribution of the first token, which may be insufficient to capture long-range sentence-level distributions that are better reflected in DNA. **We note that this comparison used Qwen2Tokenizer to align the two methods instead of using the first 4 characters, which is technically a slightly different version of PhyloLM. We also emphasize that the comparison is conducted on relation prediction, a task native to LLM DNA rather than the primary focus of PhyloLM. This does not diminish PhyloLM’s demonstrated effectiveness in its intended applications, such as benchmark performance prediction. In future work, to enable a fairer and more comprehensive comparison, we will evaluate both methods on the datasets and tasks originally used to develop PhyloLM, such as model performance prediction benchmarks. Furthermore, we will investigate more deeply how sentence-level, long-range patterns affect LLM representations.**

---

> ### Comment · Reviewer_ZryQ · 2025-11-25
> **Follow up**
>
> I thank the authors for their very complete responses throughout this discussion and I'd like to congratulate them for this very good work. I think the paper is very strong, built on original and very relevant ideas for the machine learning community, defines clearly its concepts, compares transparently to the state of the art and could deserve being highlighted at the conference. I updated my grade.
>
> A last suggestion for the authors, calling the method "DNA" may limit the reach of the paper as this is already a technical term widely used in the scientific literature. Even if the technique computes a quantity akin to DNA naming it that way could make it particularly hard for readers to find it on the internet. Most algorithms try to have a very specific and unique name to be well recognised and referenced on the internet and on arXiv (eg. among the works cited EmbedLLM, PhyloLM, ...). For example, a reader just seeing table 6 (the one with the comparison with baselines) on social media or in a presentation will easily find the PhyloLM paper by typing "PhyloLM" but will hardly find the authors' paper by typing "DNA". This does not mean the word "DNA" should disappear from the paper, on the contrary it is a great analogy but the method may require a more specific name to be recognised (and put it in the title of the paper for visibility). I leave this advice solely to the consideration of the authors—please feel free to keep the current name if you prefer it. This suggestion does not affect my assessment of the work's quality.

---

> > ### Author Response · Authors · 2025-11-26
> > **Thank You and Response to Follow-up**
> >
> > We sincerely thank Reviewer `ZryQ` for raising the score and for the insightful feedback provided throughout the discussion period, which has greatly helped us improve and strengthen the paper. We also agree that “DNA” may not be an ideal name for a specific algorithm, as it conflicts with a widely used term. We will carefully consider and adopt a more distinctive and appropriate name in the final version. Thank you again for your valuable suggestions and support.

---

### Official Review · Reviewer_vzx5 · 2025-10-30

**Soundness:** 3
**Presentation:** 2
**Contribution:** 2
**Rating:** 4
**Confidence:** 3

**Summary:**

This paper proposes LLM DNA, a low-dimensional representation of large language models obtained through random projection on model outputs. The authors formally define DNA as a bi-Lipschitz embedding, prove its existence via the Johnson-Lindenstrauss lemma, and demonstrate its utility for detecting model relationships across 305 models. Experiments show 95.7% AUC in relationship detection and reveal previously undocumented model relationships through phylogenetic tree analysis.

**Strengths:**

1. The paper addresses an interesting problem of tracing LLM evolutionary relationships through the DNA concept, demonstrating potential as a practical tool.

2. The experimental scale of 300+ models is substantial, and the discovery of false negative samples shows promising value.

**Weaknesses:**

1. The paper's core issue is presenting standard dimensionality reduction through two layers of excessive framing that obscure rather than clarify the contribution:
   - Mathematical formalization: Sections 3-4 occupy too much space establishing Hilbert space formulation, bi-Lipschitz conditions, and applying the Johnson-Lindenstrauss lemma. However, this entire framework is generic, applying identically to image classifiers, recommender systems, or any function class. Theorems 3.4-3.5 contribute no LLM-specific insights, merely restating that this is random projection with standard techniques.
   - Biological framing: The DNA terminology ("inheritance," "genetic determinism" in lines 108-115) lacks citation to biological literature supporting these characterizations. For the ML community, framing this as "low-dimensional manifold learning" would be far more transparent and align with established understanding of dimensionality reduction methods. Why is phylogenetic tree construction necessary for analyzing LLM relationships? Since the method only requires a distance matrix, are there simpler established methods that could achieve similar relationship visualization without requiring biological background knowledge?

2. Related work inadequately contextualizes the approach. The "Representation of LLMs" in Section 2 mentions model routing and ensemble learning without explaining these tasks or their connection to LLM DNA extraction, leaving unclear how this work differs from or builds upon existing representation learning methods.

3. Baseline comparisons are critically insufficient. This is fundamentally model similarity measurement, yet experiments only compare against "Random" and "Greedy" strawmen plus EmbedLLM.

Overall, I recommend the authors reduce the extensive mathematical and biological framing that appears sophisticated but contributes no novel insights, and instead devote more effort to contextualizing the work within existing literature and conducting more rigorous experiments.

**Questions:**

1. Phylogenetic trees in biology represent species-level divergence, yet your tree shows iterative model versions like Qwen2.5 and Qwen3 as separate branches. Do you think the biological evolution analogy genuinely fits the nature of LLM development, or does it create conceptual mismatches?

---

> ### Author Response · Authors · 2025-11-23
> **Author Response (Part 1)**
>
> We thank Reviewer `vzx5` for your insightful and constructive comments, which have greatly helped improve this paper. We address all concerns below, and the corresponding revisions are included in the **revised paper** (see the updated PDF with **changes highlighted in blue**). For convenience, we also paste the revised text (excluding figures and tables) in our response.
>
> ---
>
> ### Weakness 1: "The paper's core issue is presenting ..."
>
> **Response**: We appreciate the advice. In the revised paper, we have improved the presentation by:
>
> * **Simplifying the math formulation:** We moved the Hilbert-space formulation and the formal statements of the two properties to the appendix, and merged the definition and existence proof into a single Section 3. The main paper now retains only the DNA definition and the existence claims, which are necessary for problem setup and motivating Algorithm 1.
> * **Removing the biological framing:** We removed the biological motivation paragraph (original lines 108-115) and now describe the two properties in machine-learning terms in the main text.
>
> > **Section 3.1: Definition of LLM DNA**
> >
> >...
> >These lower and upper bounds in the bi-Lipschitz condition naturally guarantee two key properties of LLM DNA: (1) **Inheritance**—a lineage operation on an LLM, such as fine-tuning, produces descendants with similar DNAs; and (2) **Genetic determinism**—LLMs with similar DNAs exhibit similar functional behaviors.
>
> Then, we address your three additional questions below.
>
> **(1) "Why is phylogenetic tree construction necessary for analyzing LLM relationships?"**
>
> Phylogenetic trees provide insights that distance matrices or clustering methods cannot:
>
> 1. **Directionality and lineage.** LLMs often have real “ancestors” created through lineage operations such as fine-tuning. A phylogenetic tree explicitly represents this directional hierarchy (e.g., tracing **Llama-3-Instruct** back to **Llama-2-Chat**), whereas clustering or distance heatmaps only show proximity without ancestry.
> 2. **Evolutionary rate.** The branch lengths in Figure 5 quantify “evolutionary distance,” i.e., the magnitude of functional change along a lineage. This enables rate comparisons across families—for instance, showing that the **Qwen** family changes functionally faster than the **Llama** family—an insight that cannot be recovered from a standard distance heatmap alone.
>
> **(2) "Are there simpler established methods?"**
>
> Yes. The state-of-the-art established approach is PhyloLM [1], which similarly constructs phylogenetic-style visualizations from a distance matrix derived from token distributions. The central challenge in this line of work is defining a general, computable distance between LLMs—this is also our key distinction from PhyloLM. We therefore include PhyloLM as a strong baseline in the revised paper. Our experiments show that **LLM DNA consistently outperforms PhyloLM** on the LLM relationship prediction task (Table 1) and exhibits **better generalizability**. A key reason is that PhyloLM relies on token-level distances, making it sensitive to tokenizer choices and harder to transfer across model families with different vocabularies, whereas our DNA is **functional and tokenizer-agnostic**, enabling more robust lineage visualization and comparison. In addition, neither our method nor PhyloLM requires biological background knowledge. We only included the biological framing to aid intuition, and it has now been removed.
>
> >**Table 1: DNA LLM-relation detection test performance (mean ±standard deviation across five seeds)**
> >|Method|Acc|Prec|Rec|F1|AUC|
> >|-|-|-|-|-|-|
> >|Random |0.493±0.036 |0.515±0.034|0.516±0.042|0.516±0.036|0.492±0.036|
> >|Greedy |0.593±0.018 |0.759±0.081|0.329±0.000|0.458±0.014|0.606±0.023|
> >|PhyloLM + SVM |0.742±0.058 |0.741±0.073 |0.812±0.123|0.766±0.057|0.788±0.060|
> >|DNA (Qwen3) + SVM |0.919±0.048 |0.898±0.049 |0.957±0.072 |0.925±0.047|0.979±0.027|
> >|DNA (BGE) + SVM|**0.934±0.053** |0.905±0.068|**0.982±0.059**|**0.940±0.050** |**0.992±0.023**|
> >|DNA (MPNet) + SVM|0.932±0.048|**0.914±0.065**|0.966±0.069|0.937±0.047|0.990±0.020|
>
> **(3) Clarification of Our Contribution**
>
> Our main contribution is to bridge the theory–empirical gap in defining and evaluating a general, stable, computable, low-dimensional representation of LLMs. Although the bi-Lipschitz formulation is natural, as you noted, no prior work (e.g., PhyloLM [1], EmbedLM [2]) adopts this perspective or builds methods on it, largely because it is nontrivial to make such a formulation operational.
>
> We fill this gap by (i) defining a principled approximation of the bi-Lipschitz distance (Definition 4.1), (ii) providing a concentration guarantee (Lemma 4.2), and (iii) making the approach practical with Algorithm 1. Most importantly, we show that this theoretically grounded method is also simple and effective in practice: across large-scale experiments on 300+ LLMs, our DNA representation outperforms PhyloLM [1] and EmbedLM [2].
>
> ---

---

> ### Author Response · Authors · 2025-11-23
> **Author Response (Part 2)**
>
> ### Weakness 2: "Related work inadequately contextualizes the approach."
>
> **Response**: We revised the paper to include a more detailed explanation of model routing and ensemble-learning methods. These approaches are not applicable to DNA extraction because they rely on supervised training for specific downstream tasks, and thus their learned representations are task-dependent and do not generalize across tasks.
>
>
>
> > **Section 2: Related Work**
> >
> >  **Representation of LLMs.** This direction manages large LLM collections by encoding each LLM into a representation. Most methods are task-specific: Ong et al. (2025); Wang et al. (2024) learn representations for model routing, while Huang et al. (2024); Fang et al. (2024) encode models for ensemble learning. For instance, HybridLLM (Ding et al., 2024) trains a binary “easy-hard” classifier to route queries to a small or a larger model. RouteLLM (Ong et al., 2025) learns a supervised router that selects between stronger and weaker models based on query features. MetaLLM (Nguyen et al., 2024) formulates routing as a multi-armed bandit, adaptively trading off expected answer quality against invocation cost via online feedback. LLM-ensemble (Fang et al., 2024) learns the weights for different LLMs to predict a specifc attribute value. **These task-specific approaches rely on supervised training for particular tasks, making their learned representations not generalizable across tasks.** Zhuang et al. (2025) propose EmbedLLM, the first compact embedding supporting multiple tasks (e.g., model routing and benchmark prediction), but it is defined over a fixed model set; adding models requires retraining and can shift the learned embeddings. Yax et al. (2025) propose PhyloLM, which constructs a phylogenetic tree using a novel distance between LLMs. However, this distance is token-based and does not generalize well to heterogeneous LLMs. In contrast, our LLM DNA is a stable and generalizable intrinsic property extractable independently for each LLM.
>
> ---
>
> ### Weakness 3: "Baseline comparisons are insufficient."
>
> **Response**: We include a comparison with the state-of-the-art approach PhyloLM [1] as suggested by Reviewer `ZryQ`, which similarly constructs phylogenetic-style visualizations from a distance matrix derived from token distributions. Our experiments show that LLM DNA consistently outperforms PhyloLM on the LLM relationship prediction task (Table 1) and demonstrates better generalizability. Please see the **updated Table 1** in the revised paper, as well as our **response to “Weakness 1,”** for further details.

---

> ### Author Response · Authors · 2025-11-23
> **Author Response (Part 3)**
>
> ---
>
> ### Question 1: "Phylogenetic trees in biology represent species-level divergence, yet your tree shows iterative model versions like Qwen2.5 and Qwen3 as separate branches. Do you think the biological evolution analogy genuinely fits the nature of LLM development, or does it create conceptual mismatches?"
>
>
> **Response**: Qwen2.5 and Qwen3 appearing as separate (but close) branches is expected, because Qwen3 is **not trained directly** on top of Qwen2.5. According to the Qwen3 technical report [3], Qwen3 is pre-trained on 36 trillion tokens, explicitly described as “*twice as many pre-training tokens*” as Qwen2.5. This implies that Qwen3 is trained from a **different base model with substantially more pre-training**, rather than being an incremental continuation of Qwen2.5. Our phylogenetic tree in Figure 5 reflects this: the mean parent–child distance for Qwen2.5 is 0.038, while for Qwen3 it is 0.067—**approximately twice** as large—matching the report’s description.
>
> More broadly, we use the biological Neighbor-Joining (NJ) algorithm not because LLM development follows biological evolution, but because NJ solves the same underlying mathematical problem. NJ reconstructs a tree from pairwise distances by minimizing the total branch length (the “minimum evolution” objective in biology). In our setting, branch length corresponds to the magnitude of **functional behavioral shift** between model checkpoints. The analogy is therefore methodological rather than ontological: we borrow a well-established distance-to-tree reconstruction tool to visualize directional lineage in functional space.
>
> The minimum-evolution principle is reasonable for LLMs due to explicit engineering constraints, not biological natural selection. First, **catastrophic-forgetting mitigation** in fine-tuning typically constrains optimization (often via regularization such as KL penalties) to keep descendants close to their parents in function space, which encourages minimal functional drift. Second, **parsimony in high-dimensional function space** makes accidental convergence of independently trained models to nearly identical behaviors statistically unlikely; thus, the tree topology minimizing total branch length (NJ algorithm) is a robust way to infer shared lineage from distances. Taken together, the biological framing is not required for correctness, and the tree does not assume a literal evolutionary process—it is a principled visualization of lineage under functional distances.
>
>
>
> ### **References**
>
> [1] Yax, Nicolas, Pierre-Yves Oudeyer, and Stefano Palminteri. "Phylolm: Inferring the phylogeny of large language models and predicting their performances in benchmarks." ICLR 2025.
>
> [2] Zhuang, Richard, et al. "EmbedLLM: Learning compact representations of large language models." ICLR 2025.
>
> [3] Yang, An, et al. "Qwen3 technical report." arXiv preprint arXiv:2505.09388 (2025).

---

> > ### Comment · Reviewer_vzx5 · 2025-11-23
> >
> > Thanks for the authors' strong response. I appreciate the key idea of the paper. It's somewhat interesting for me. The authors' response addressed my core concern through simplifying the overly complex presentation and the unfounded biological definitions. Besides, the added baseline looks like the results are good. I am happy to raise my score.
> >
> > Finally, I have an open question: do the authors think that the evolution process of LLMs can be understood with biological evolution theory (e.g.  natural selection)? How to understand the evolution of LLMs from a high-level perspective?

---

> > > ### Author Response · Authors · 2025-11-23
> > > **Response to Follow-up Open Question**
> > >
> > > Thank you for your appreciation, interest in our work, and for raising the score. We address your open question below.
> > >
> > > Our view is that many tools, theories, and methodologies from biological evolution can help study LLM evolution, but they often need adaptation. This is because the two processes share several high-level characteristics:
> > >
> > > * **Evolution tends to minimize drift.** In biology, mutation is relatively rare, so descendants typically stay close to their ancestors. In LLMs, catastrophic-forgetting mitigation during fine-tuning (often via regularization such as KL penalties) similarly constrains optimization to keep descendants close to their parents in function space, encouraging minimal functional drift.
> > >
> > > * **Evolution is shaped by selection pressures.** In biology, better-adapted species tend to survive and diversify. In LLMs, stronger base models (e.g., Qwen) attract many descendants through fine-tuning or distillation and thus continue to branch and develop, while weaker lineages gradually stop evolving as they lose adoption (e.g., Flan-T5 and the encoder–decoder branch).
> > >
> > > * **Independent convergence to the same “species” is rare.** In biology, mutations are undirected, making accidental convergence extremely unlikely. In LLMs, parsimony in high-dimensional function space likewise makes independent training leading to nearly identical behaviors statistically improbable; convergent evolution may align certain traits, but not the entire DNA.
> > >
> > > Given these commonalities, we believe phylogenetic perspectives are valuable for understanding LLM evolution. At the same time, directly importing biological models is not always appropriate. For instance, many advanced phylogenetic tree-building algorithms assume specific mutation structures and rates (e.g., over four nucleotide types--ATCG), whereas LLM DNA is far richer and evolves under different mechanisms. Developing evolution models tailored to LLMs is therefore an interesting but challenging direction, and we plan to pursue it in future work.

---

### Official Review · Reviewer_Jnw8 · 2025-10-31

**Soundness:** 3
**Presentation:** 3
**Contribution:** 3
**Rating:** 6
**Confidence:** 3

**Summary:**

This paper introduces LLM DNA, a formal framework to identify and trace the evolutionary relationships of large language models (LLMs). The authors address the challenge that with millions of LLMs existing, their origins through fine-tuning, distillation, or adaptation are often unclear.

**Strengths:**

-one of the paper's  strengths is its rmathematical grounding. It doesn't just propose a heuristic; it formally defines LLM DNA using the Bi-Lipschitz condition and uses the Johnson-Lindenstrauss (JL) Lemma to prove its existence.

-Unlike methods that require retraining to compare models, the LLM DNA extraction pipeline is fixed. Once the random projection matrix $A$ and prompt set $\mathcal{S}_t$ are defined, any new model can have its DNA extracted independently4. This makes the system highly scalable and durable.

-The method does not rely on model internals like architecture, weights, or tokenizers. Since it only evaluates text outputs , it works on black-box API models (like GPT-4) just as easily as open-source models, and it can compare models of completely different types

**Weaknesses:**

-While the Mantel test shows stability for general datasets, the DNA is still a snapshot of the model's behavior on only $t=600$ prompts. If this prompt set were narrowly-scoped (e.g., all prompts were Python coding problems), the resulting DNA would likely represent a "coding DNA" rather than the model's general functional behavior.

-.The DNA measures functional similarity (i.e., "do these models behave alike?"). If two models from different companies were trained independently but happened to achieve a very similar functional state (convergent evolution), their DNAs would be nearly identical.

**Questions:**

-the paper sets the final DNA dimension $L=128$. How was this number chosen? How sensitive is the DNA's quality to this choice? If $L=32$, would the system lose the resolution to distinguish highly similar models (e.g., Llama-3-8B vs. Llama-3-8B-Instruct)?

-How sensitive is the method to minor fine-tuning? If I fine-tune Llama 3 on just 1,000 new samples, how much will its DNA $\tau_{f_2}$ change from the original $\tau_{f_1}$? Is the DNA distance $d_\tau$ predictably correlated with the amount or domain of fine-tuning data?

-The study includes models from 100M to 70B parameters. Is the method equally effective across this entire range? Does using a powerful embedding model $\phi$ on the (often lower-quality) outputs of a 100M model "over-beautify" its functional representation, making its DNA seem more capable than it is?

---

> ### Author Response · Authors · 2025-11-23
> **Author Response (Part 1)**
>
> We thank Reviewer `Jnw8` for your insightful and constructive comments, which have greatly helped improve this paper. We address all concerns below, and the corresponding revisions are included in the **revised paper** (see the updated PDF with **changes highlighted in blue**). For convenience, we also paste the revised text (excluding figures and tables) in our response.
>
> ---
>
> ### Weakness 1: "DNA representing a specific scope (e.g., "coding DNA") rather than general behavior."
>
> **Response**: **DNA is insensitive to the evaluated datasets**. We confirm this by an additional experiment in the **Appendix C.7 of revised paper**. The results show that even replacing the standard benchmark inputs with randomly generated data does not degrade the quality of DNA. This is because DNA analyses (e.g., relationship prediction) target *relative relationships* among LLMs rather than absolute accuracy.
>
>
> >**Appendix C.7: DNA Extracted from Random Data**
> > This subsection demonstrates the strong robustness of LLM DNA: it preserves distinguishability even when the evaluation inputs are random. Specifically, we use `wonderwords.RandomWord` to generate 600 samples (matching our main evaluation: 6 datasets with 100 samples each). Each sample is a 100-word sentence. We run inference with each LLM on this dataset, collect their outputs, and encode them using Qwen3-8B-Embedding, identical to the main DNA extraction process. We then perform the relation prediction task using these DNAs. An example slice of the random input data is:
> >
> > ```
> > disdain chapel intention gymnast activation nymph prickly ... scout cutting x-ray shack codpiece hypothesis endothelium masonry
> > ```
> >
> > The results are reported in Table 4. Interestingly, the **random-data setting even slightly outperforms the standard benchmark**. This highlights that the DNA extraction method is largely insensitive to input-dataset bias. The key intuition is that DNA analyses (e.g., relationship prediction) target *relative relationships* among LLMs rather than absolute accuracy. Different datasets act as different “views” of the same model set: a suboptimal view may obscure some relations, but it is unlikely to fabricate spurious ones—for instance, it will not make two genuinely similar models appear unrelated. This observation also aligns with the stability we see under random projection.
> >
> > **Table 4: DNA (Qwen3) relation prediction performance (DNA extracted from random inputs)**
> >
> > |Method|Accuracy|Precision|Recall|F1|AUC|
> > |-|-|-|-|-|-|
> > |PhyloLM + SVM| 0.742 ± 0.058|0.741 ± 0.073|0.812 ± 0.123|0.766 ± 0.057|0.788 ± 0.060|
> > |DNA + SVM| *0.919 ± 0.048*|*0.898 ± 0.049*|*0.957 ± 0.072*|*0.925 ± 0.047*|*0.979 ± 0.027*|
> > |DNA (Rand) + SVM|**0.949 ± 0.048**|**0.931 ± 0.053**|**0.977 ± 0.062**|**0.952 ± 0.048** | **0.987 ± 0.038** |
>
> ---
>
> ### Weakness 2: "... If two models from different companies were trained independently but happened to achieve a very similar functional state (convergent evolution), their DNAs would be nearly identical."
>
>
> **Response**: Convergent evolution may indeed occur, but it would make only certain subsequence (traits) similar rather than render the entire DNA nearly identical. This is because **LLM DNA is sensitive to slight fine-tuning**: even fine-tuning with just 10 samples from an external dataset produces a clear shift in DNA relative to the original model. We validate this in the revised paper with a **new experiment in Section 5.6**, with results shown in **Figure 6**. Therefore, independently obtaining the same full DNA would be extremely unlikely—analogous to two species independently evolving into humans.
>
>
> > **Section 5.6: Effect of Fine-tuning on the DNA Values**
> >
> >This subsection quantifies how fine-tuning changes DNA values. Starting from `Llama3-8B-Instruct`, we fully fine-tune all parameters on subsets of the NVIDIA `OpenMathInstruct-2` dataset (Toshniwal et al., 2024) with sizes $\\{10, 100, 1000, 10000\\}$. Since `Llama3` was released before `OpenMathInstruct-2`, this dataset appears in neither its pretraining corpus nor our DNA extraction set. Figure 6 compares the DNAs of the original and fine-tuned models and reports their $L_2$ distances to the original DNA.
> >
> >We draw two conclusions. First, **the DNA distance increases with the amount of fine-tuned data, showing a monotonic but nonlinear trend**. Specifically, the distance rises sharply after fine-tuning on a tiny external subset (e.g., the first 10 samples causes $L_2=0.69$) and then grows more gradually (up to 0.80 with 10,000 samples) as additional samples are learned. Second, **the global DNA structure remains stable**: most subsequences preserve their dark/light patterns with only minor variation, suggesting that fine-tuning alters local traits while leaving inherited, intrinsic behaviors largely intact.
>
>
> ---

---

> ### Author Response · Authors · 2025-11-23
> **Author Response (Part 2)**
>
> ### Question 1: Choice of Dimension $L$ and Resolution
>
> **Response**: DNA quality improves as the DNA dimension $L$ (denoted as $d$ by the reviewer) increases, and it plateaus beyond a convergence point ($L=128$ in our relation prediction experiments). In the revised paper, we include a **new ablation study in Appendix C.5**, with results shown in **Figure 10**, to demonstrate this trend.
>
>
> > **C.5 Ablation Study of the DNA Dimension $L$**
> >
> > This ablation study examines how the DNA dimension $L$ affects DNA quality. We vary $L \in \\{4, 32, 128, 1024, 4096\\}$ and evaluate each setting on the same relationship prediction task used in Table 1. The results are shown in Figure 10. **DNA quality improves as $L$ increases and converges around $L=128$**. Very low dimensions (e.g., $L=4$) underperform because they discard structural information from the original high-dimensional space. Thus, selecting $L$ involves a **tradeoff between quality and efficiency**: larger dimensions preserve more structural detail and improve accuracy, but incur increasing downstream computation and storage costs; smaller dimensions are cheaper and faster, but can overly compress the space and lose critical structure, leading to degraded DNA quality. We therefore use $L=128$ in all experiments. Note that this choice is specific to our current evaluation scale of approximately 300 LLMs; for substantially larger model sets, a larger $L$ may be required.
>
> ---
>
> ### Question 2: Sensitivity to Minor Fine-tuning
>
> **Response**: The DNA distance increases with the amount of fine-tuned data, showing a monotonic but nonlinear trend. Specifically, the distance rises sharply after fine-tuning on a very small external subset (e.g., the first 10 samples) and then grows more gradually as additional samples are learned. We demonstrate this behavior with a **new experiment in Section 5.6**, with results shown in **Figure 6** of the revised paper.
>
> See our response to **Weakness 2** for the quoted revised text.
>
> ---
>
> ### Question 3: "Is the method equally effective across this entire range?"
>
>
> **Response**: In the revised paper, we demonstrate that (1) DNA quality is statistically independent of LLM size; and (2) DNA quality does not depend on embedding model size.
>
> ### **(1) DNA quality is statistically independent of LLM size**
>
> This is demonstrated by a **new statistical experiment in Appendix C.6**, with results shown in **Table 4**.
>
> > **Appendix C.6: Effect of LLM's Parameter Size on DNA Quality**
> >
> > This subsection investigates whether the performance of LLM DNA is biased by model scale—specifically, whether the method effectiveness varies significantly between small and large models. To rigorously test this, we conduct a post-hoc error analysis completely separate from the DNA classification pipeline. The DNA classifier first predicts relationships using only functional embeddings, without access to metadata such as parameter counts. After fixing these predictions, we calculate the *cumulative parameter size* for every test pair (the sum of parameters of both models). We then compute the *Point-Biserial correlation* ($r_{pb}$) and fit an independent logistic regression model to determine if this cumulative size can predict the *correctness* of the DNA classifier's output (1 for correct, 0 for incorrect). Table 4 reports the performance metrics alongside the correlation coefficient and the $p$-value of the size coefficient derived from the regression analysis.
> >
> > The results yield two key observations. First, **DNA quality is statistically independent of LLM size**: across all DNA-based methods, the Point-Biserial correlations are negligible (ranging from -0.006 to 0.023), and the $p$-values for the size coefficient are substantial (ranging from 0.63 to 0.69), well above the standard significance threshold of 0.05. This indicates that the method's ability to detect evolutionary relationships does not degrade for larger models nor improve for smaller ones. Second, **robustness is consistent across different encoders regardless of their own size**: the lack of size bias (high $p$-value) remains constant across the large Qwen3 (8B) encoder and the much smaller BGE (0.3B) and MPNet (0.1B) encoders. This suggests that LLM DNA captures intrinsic functional behaviors that persist across model scales, rather than relying on complexity metrics associated with parameter count.
> >
> >**Table 4: Point-Biserial correlation ($r_{pb}$) and statistical significance ($p$-value) between LLM size and relation prediction accuracy**
> >
> >| Method|Acc|F1|AUC|$r_{pb}$|$p$|
> >|-|-|-|- |-|-|
> >|PhyloLM|0.742±0.058 |0.766±0.057|0.788±0.060|−0.097±0.195|0.520|
> >|DNA–Qwen3|0.919±0.048| 0.925±0.047|0.979±0.027|0.025±0.123|0.635|
> >|DNA–BGE|0.934±0.053|0.940±0.050|0.992±0.023|0.013±0.125|0.655|
> >|DNA–MPNet|0.932±0.048|0.937±0.047|0.990±0.020|0.023±0.122|0.690|

---

> ### Author Response · Authors · 2025-11-23
> **Author Response (Part 3)**
>
> ### **(2) DNA quality does not depend on embedding model size**
>
> This is confirmed with a **new ablation studies in Appendix C.3** of the revised paper, with results shown in **Figure 8**. This is further supported by the **new results in Table 1** showing that simple sentence-embedding models can achieve strong relation prediction performance.
>
>
> > **Appendix C.3: Ablation Study of Bottleneck Embedding Model**
> > This ablation study evaluates the effect of the bottleneck embedding model on the computed DNA. In addition to Qwen-8B-Embedding, we compute DNA using two popular sentence-embedding models from different organizations and with different sizes: `all-mpnet-base-v2` (0.1B parameters) and `bge-large-en-v1.5` (0.3B parameters). We then perform a Mantel test—a standard statistical test for correlating two distance matrices—pairwise across the three DNA variants. Specifically, we sample 10,000 random DNA pairs, compute their distances, and report the Spearman correlation and p-value in Figure 8. **All pairs of embedding models exhibit strong agreement, with high correlations ($r>0.75$) and strong statistical significance ($p<0.0001$)**. These results indicate that the choice of embedding model has marginal impact on the overall DNA structure; even a small (0.1B) embedding model yields comparable DNA.
>
>
> > **Section 6.2: Detect LLM Relationships**
> > From Table 1, three observations emerge. ... Third, **DNA quality is robust to the choice of sentence-embedding model**. Using BGE (0.3B) and MPNet (0.1B), despite their much smaller parameter sizes, yields comparable—and in this relation prediction task, even better—performance.
> >
> >**Table 1: DNA LLM-relation detection test performance (mean ±standard deviation across five seeds)**
> >|Method|Acc|Prec|Rec|F1|AUC|
> >|:-|-:|-:|-:|-:|-:|
> >|Random |0.493±0.036 |0.515±0.034|0.516±0.042|0.516±0.036|0.492±0.036|
> >|Greedy |0.593±0.018 |0.759±0.081|0.329±0.000|0.458±0.014|0.606±0.023|
> >|PhyloLM + SVM |0.742±0.058 |0.741±0.073 |0.812±0.123|0.766±0.057|0.788±0.060|
> >|DNA (Qwen3) + SVM |0.919±0.048 |0.898±0.049 |0.957±0.072 |0.925±0.047|0.979±0.027|
> >|DNA (BGE) + SVM|**0.934±0.053** |0.905±0.068 |**0.982±0.059**|**0.940±0.050** |**0.992±0.023** |
> > | DNA (MPNet) + SVM|0.932±0.048|**0.914±0.065**|0.966±0.069| 0.937±0.047|0.990±0.020|

---

### Official Review · Reviewer_5DVv · 2025-11-04

**Soundness:** 3
**Presentation:** 3
**Contribution:** 3
**Rating:** 6
**Confidence:** 2

**Summary:**

The paper proposes LLM DNA, a mathematically grounded, low-dimensional representation intended to capture an LLM’s functional behavior. The approach: (i) define DNA via a bi-Lipschitz map over a space of LLM functions, (ii) justify existence with a Johnson–Lindenstrauss (JL) argument, and (iii) provide a training-free pipeline. Empirically, DNAs cluster related models and support simple routing.

**Strengths:**

- The bi-Lipschitz view + JL existence yields a clean, theory-backed rationale rather than an ad-hoc embedding and is well principled.
- The training-free pipeline is straightforward, practical, and appears to work across heterogeneous and API-only models.
- The high-level theory (bi-Lipschitz definition, JL-style existence, and concentration arguments) appears reasonable for finite settings.
- The method is agnostic to architecture and tokenizers, and doesn't require rebuilding an embedding space when adding LLMs to the evaluation set. This means it can be extended to the closed-source LLMs and can be used to uncover undocumented relationships between models (p7-8, Appendix C.2), which is an advantage over EmbedLLM.

**Weaknesses:**

- "DNA" quality likely hinges on the chosen sentence-embedding backbone and its dimensionality. Robustness to encoder swaps (and multilingual/domain shifts) is not yet demonstrated.
- There is an insignificant difference in performance for the model routing test between the standard baselines.
- The motivation for the benefits that this type of representation unlocks (namely, phylogenetic analysis on models) is not sufficient. The authors show and discuss a phylogenetic tree, but discuss the genetic evolutionary rate without drawing the connection to its significance in this context and seem to be borrowing from authority of biological genetic evolutionary analysis.
- The mathematical guarantees are for finite sets and bounded inputs; it’s unclear how distortion and stability behave as input lengths or the set of models grow. Please discuss the unbounded-input case and evolving model pools.

**Questions:**

- How can the trees be useful in practice?
- Since there is only a small performance difference, are there other benefits to using this method over others?
- How do results change with different sentence-embedding models and embedding dimensionalities?
- What is the empirical trade-off between projection dimension d, distortion of pairwise distances, and downstream tasks (lineage/routing)?

---

> ### Author Response · Authors · 2025-11-23
> **Author Response (Part 1)**
>
> We thank Reviewer `5DVv` for your insightful and constructive comments, which have greatly helped improve this paper. We address all concerns below, and the corresponding revisions are included in the **revised paper** (see the updated PDF with **changes highlighted in blue**). For convenience, we also paste the revised text (excluding figures and tables) in our response.
>
> ---
>
> ### Weakness 1: "'DNA' quality likely hinges on the chosen sentence-embedding backbone and its dimensionality. Robustness to encoder swaps (and multilingual/domain shifts) is not yet demonstrated."
>
> **Response**: DNA quality is **robust to (1) the choice of sentence-embedding backbone, (2) the embedding dimension, and (3) the language/domain of the evaluation dataset**. We confirm this with three new experiments. For (1), **Figures 8 in Appendix C.3** and new results in **Table 1** show that even simple sentence-embedding models achieve strong relation-prediction performance. For (2), **Figures 9 in Appendix C.4** indicate that the output dimension $p$ has only a marginal impact on performance. For (3), **Table 5 in Appendix C.7** shows that DNA extracted from random data even slightly outperforms the standard benchmark.
>
>
> > **Appendix C.3: Ablation Study of Bottleneck Embedding Model**
> >
> > This ablation study evaluates the effect of the bottleneck embedding model on the computed DNA. In addition to Qwen-8B-Embedding, we compute DNA using two popular sentence-embedding models from different organizations and with different sizes: `all-mpnet-base-v2` (0.1B parameters) and `bge-large-en-v1.5` (0.3B parameters). We then perform a Mantel test—a standard statistical test for correlating two distance matrices—pairwise across the three DNA variants. Specifically, we sample 10,000 random DNA pairs, compute their distances, and report the Spearman correlation and p-value in Figure 8. **All pairs of embedding models exhibit strong agreement, with high correlations ($r>0.75$) and strong statistical significance ($p<0.0001$)**. These results indicate that the choice of embedding model has marginal impact on the overall DNA structure; even a small (0.1B) embedding model yields comparable DNA.
>
>
> > **Appendix C.4: Ablation Study of the Embedding Model’s Output Dimension $p$**
> > This ablation study investigates the effect of the pre-aggregation embedding dimension $p$ from Qwen3-8B-Embedding on DNA quality. We vary $p \in \\{4, 64, 128, 1024, 4096\\}$ and evaluate each setting on the same relationship prediction task used in Table 1. The results are shown in Figure 9. **Across all metrics, performance remains largely unchanged, indicating that $p$ has only a marginal impact.** This suggests that DNA quality is not primarily driven by the embedding model’s output dimension, but by the subsequent aggregation process. It also reinforces the **observation in Table 1 that even simple sentence-embedding models can achieve strong relation prediction performance**.
>
>
> > **Section 5.2: Detect LLM Relationships**
> >
> > From Table 1, three observations emerge. ... Third, **DNA quality is robust to the choice of sentence-embedding model**. Using BGE (0.3B) and MPNet (0.1B), despite their much smaller parameter sizes, yields comparable—and in this relation prediction task, even better—performance.
> >**Table 1: DNA LLM-relation detection test performance (mean ±standard deviation across five seeds)**
> >
> >|Method|Acc|Prec|Rec|F1|AUC|
> >|:-|-:|-:|-:|-:|-:|
> >|Random |0.493±0.036 |0.515±0.034|0.516±0.042|0.516±0.036|0.492±0.036|
> >|Greedy |0.593±0.018 |0.759±0.081|0.329±0.000|0.458±0.014|0.606±0.023|
> >|PhyloLM + SVM |0.742±0.058 |0.741±0.073 |0.812±0.123|0.766±0.057|0.788±0.060|
> >|DNA (Qwen3) + SVM |0.919±0.048 |0.898±0.049 |0.957±0.072 |0.925±0.047|0.979±0.027|
> >|DNA (BGE) + SVM|**0.934±0.053** |0.905±0.068|**0.982±0.059**|**0.940±0.050** |**0.992±0.023** |
> > | DNA (MPNet) + SVM|0.932±0.048|**0.914±0.065**|0.966±0.069|0.937±0.047|0.990±0.020|

---

> ### Author Response · Authors · 2025-11-23
> **Author Response (Part 2)**
>
> >**Appendix C.7: DNA Extracted from Random Data**
> >
> > This subsection demonstrates the strong robustness of LLM DNA: it preserves distinguishability even when the evaluation inputs are random. Specifically, we use `wonderwords.RandomWord` to generate 600 samples (matching our main evaluation: 6 datasets with 100 samples each). Each sample is a 100-word sentence. We run inference with each LLM on this dataset, collect their outputs, and encode them using Qwen3-8B-Embedding, identical to the main DNA extraction process. We then perform the relation prediction task using these DNAs. An example slice of the random input data is:
> >
> > ```text
> > disdain chapel intention gymnast activation nymph prickly chalk sad method historical ... omniscient pressurization surfboard infrastructure parachute ferryboat download aim scout cutting x-ray shack codpiece hypothesis endothelium masonry
> > ```
> >
> > The results are reported in Table 4. Interestingly, the **random-data setting even slightly outperforms the standard benchmark**. This highlights that the DNA extraction method is largely insensitive to input-dataset bias. The key intuition is that DNA analyses (e.g., relationship prediction) target *relative relationships* among LLMs rather than absolute accuracy. Different datasets act as different “views” of the same model set: a suboptimal view may obscure some relations, but it is unlikely to fabricate spurious ones—for instance, it will not make two genuinely similar models appear unrelated. This observation also aligns with the stability we see under random projection.
> >
> > **Table 4: DNA (Qwen3) relation prediction performance (DNA extracted from random inputs)**
> >
> > | Method | Accuracy | Precision | Recall | F1 | AUC|
> > | - | - | - | - | - | - |
> > | PhyloLM + SVM| 0.742 ± 0.058|0.741 ± 0.073|0.812 ± 0.123|0.766 ± 0.057|0.788 ± 0.060|
> > | DNA + SVM| *0.919 ± 0.048*   |*0.898 ± 0.049*|*0.957 ± 0.072*|*0.925 ± 0.047*|*0.979 ± 0.027*|
> > | DNA (Rand) + SVM |**0.949 ± 0.048**|**0.931 ± 0.053**|**0.977 ± 0.062**|**0.952 ± 0.048** | **0.987 ± 0.038** |
>
> ---
>
> ### Weakness 2: "There is an insignificant difference in performance for the model routing test between the standard baselines."
>
> **Response:** We argue that achieving parity with EmbedLLM is a significant strength, not a weakness, given the **methodological differences**:
>
> 1.  **Zero-Shot vs. Supervised:** EmbedLLM is supervised and **explicitly trained on this routing dataset**. Our method is **training-free** and never sees the task data. Matching the performance of a supervised baseline using only intrinsic DNA validates the high quality of our representations.
> 2.  **Generalizability:** EmbedLLM requires retraining to accommodate any new model. In contrast, LLM DNA can be extracted for **arbitrary new models independently** and used immediately, offering superior scalability and real-world utility.
>
> We have revised Section 5.3 to emphasize these points:
> >This result is notable because EmbedLLM explicitly learns representations on this routing dataset, whereas LLM DNA is training-free and task-agnostic. Achieving parity without task-specific optimization validates the quality of DNA and highlights its superior scalability: unlike EmbedLLM, our model supports arbitrary new models without retraining.
>
> ---
>
> ### Weakness 3: "The motivation for the benefits that this type of representation unlocks (namely, phylogenetic analysis on models) is not sufficient. ..."
>
> **Response**: We have revised the manuscript to highlight three concrete applications that can benefit LLM ecosystem in **Appendix D**:
>
> >**Appendix D: Broader Impact**
> >
> >The utility of the phylogenetic tree extends beyond historical visualization, offering actionable insights into three key aspects of model management.
> >
> >**License Governance.** As the number of models grow rapidly, non-compliance with model licenses is becoming a critical concern (Duan et al., 2025). Such behaviors include fine-tuning or distillation from a model without giving proper credit or respecting license terms. For example, Yoon et al. (2025) allege that Huawei's Pangu-Pro-MoE is derived from Qwen-2.5-14B, a claim later denied by Huawei. Existing license-analysis tools such as ModelGo (Duan et al., 2024) primarily rely on metadata or explicit claims. In contrast, our phylogenetic analysis provides **functional indication of lineage**. For instance, our tree clusters Microsoft's `orca-2-13b` and LMSYS's `vicuna-7b-v1.1` tightly, identifying a shared lineage that is vaguely mentioned in model cards but not explicitly documented in Hugging Face. The phylogenetic tree offers a data-driven indication for flagging potential license-violation cases and strengthening license governance in the LLM ecosystem.

---

> ### Author Response · Authors · 2025-11-23
> **Author Response (Part 3)**
>
> >**Model Selection.** Our framework facilitates informed model selection by analyzing the interplay between functional inheritance and iteration velocity. First, the phylogenetic lineage allows users to **identify families where specialized capabilities**—such as reasoning patterns (Huang & Chang, 2023) or domain knowledge (Wu et al., 2025)—are likely transferred down from ancestor models, ensuring the retention of desired baseline traits. Second, the **“evolutionary rate” (branch length) serves as a proxy for *Iteration Velocity***, quantifying the magnitude of functional change between versions. This reveals a fundamental **trade-off between novelty and stability**: while a high iteration velocity offers significant behavioral shifts advantageous for *experimental use cases*, models with lower velocity (higher backward compatibility) are often preferable for *production environments*—particularly multi-agent systems—to minimize the risk of breaking established agent workflows or prompt dependencies.
> >
> >**Risk Mitigation.** Finally, the phylogenetic tree facilitates proactive **risk mitigation by tracing the propagation of safety risks**. Our findings on inheritance suggest that vulnerabilities—such as implanted backdoors (Yang et al., 2024) or jailbreak susceptibility (Russinovich et al., 2025)—are probably transferred down model lineages. Consequently, the tree allows auditors to propagate known risk profiles from ancestor models to their descendants. If a parent model is flagged as unsafe, its descendants identified by the tree can be immediately prioritized for safety interventions. This enables auditors to mitigate deployment risks efficiently, focusing resources on high-risk lineages without requiring exhaustive ground-up testing for every new model iteration.
>
>
>
> ---
>
> ### Weakness 4: "The mathematical guarantees are for finite sets and bounded inputs; it’s unclear how distortion and stability behave as input lengths or the set of models grow. Please discuss the unbounded-input case and evolving model pools."
>
> **Response**: We emphasize that our theoretical prerequisite of "finiteness" does not imply “small” or “limited.” On the contrary, the sets involved can be **arbitrarily large**. Thus, the “finite set” and “bounded input” conditions are not restrictive assumptions, but rather faithful reflections of computational reality.
>
> 1.  **Bounded Inputs are a Physical Fact:** The input space is naturally finite because the token vocabulary $\Sigma$ is finite and the length of generated text—even if up to, say, one trillian tokens—is still bounded in any real system. Truly infinite-length text would require infinite time to read and process, which is not meaningful in practice. This is impracticibility is also supported by multiple theoretical studies [1,2].
> 2.  **Scalability to Evolving Pools:** Our existence proof (Theorem 3.1) shows the required DNA dimension $L$ scales logarithmically with the number of models ($L = O(\log K)$). This ensures robust scalability: an exponential increase in the number of models requires only a linear increase in DNA dimension to maintain the same distortion bounds.
> 3.  **Stability:** Because our projection matrix $A$ is random and fixed (Corollary 3.2), the DNA extraction is independent for each model. Adding new models to the pool does not alter the DNA or stability of existing models, unlike learned embeddings which would require retraining.
>
> We clarify this point in **Section 3.1** of the revised paper.
>
> >For the theoretical analysis, we assume the input length is finite, i.e., bounded by a constant $m$. Notably, “finiteness” does not imply smallness; the sets involved may still be arbitrarily large. Meanwhile, our practical stochastic-approximation pipeline in Section~\ref{sec:dna-find} does not depend on any particular value of $m$.
>
>
> ---
>
> ### Question 1: "How can the trees be useful in practice?"*
>
> **Response**: See our response to **Weakness 3**.
>
> ---
>
> ### Question 2: "Since there is only a small performance difference, are there other benefits to using this method over others?"*
>
> **Response**: In the revised paper, we show that DNA significantly outperforms PhyloLM [3] on relation prediction (Table 1). For EmbedLLM, please see our response to **Weakness 2**.
>
> ---
>
> ### Question 3: "How do results change with different sentence-embedding models and embedding dimensionalities?"
>
> **Response**: See our response to **Weakness 1**.

---

> ### Author Response · Authors · 2025-11-23
> **Author Response (Part 4)**
>
> ---
>
> ### Question 4: "What is the empirical trade-off between projection dimension d, distortion of pairwise distances, and downstream tasks (lineage/routing)?"
>
> **Response**: DNA quality improves as the DNA dimension $L$ (denoted as $d$ by the reviewer) increases, and it plateaus beyond a convergence point ($L=128$ in our relation prediction experiments). In the revised paper, we include a **new ablation study in Appendix C.5**, with results shown in **Figure 10**, to demonstrate this trend.
>
>
> > **C.5 Ablation Study of the DNA Dimension $L$**
> >
> > This ablation study examines how the DNA dimension $L$ affects DNA quality. We vary $L \in \\{4, 32, 128, 1024, 4096\\}$ and evaluate each setting on the same relationship prediction task used in Table 1. The results are shown in Figure 10. **DNA quality improves as $L$ increases and converges around $L=128$**. Very low dimensions (e.g., $L=4$) underperform because they discard structural information from the original high-dimensional space. Thus, selecting $L$ involves a **tradeoff between quality and efficiency**: larger dimensions preserve more structural detail and improve accuracy, but incur increasing downstream computation and storage costs; smaller dimensions are cheaper and faster, but can overly compress the space and lose critical structure, leading to degraded DNA quality. We therefore use $L=128$ in all experiments. Note that this choice is specific to our current evaluation scale of approximately 300 LLMs; for substantially larger model sets, a larger $L$ may be required.
>
>
>
> ### **References**
>
> [1] Mohsin et al. On the Fundamental Limits of LLMs at Scale, arXiv preprint arXiv:2511.12869 (2025).
>
> [2] Shi, Jingzhe, et al. "Explaining context length scaling and bounds for language models." arXiv preprint arXiv:2502.01481 (2025).
>
> [3] Yax, Nicolas, Pierre-Yves Oudeyer, and Stefano Palminteri. "Phylolm: Inferring the phylogeny of large language models and predicting their performances in benchmarks." ICLR 2025.

---

### Author Response · Authors · 2025-12-01
**Discussion Summary**

## **Executive Summary: Strong Positive Consensus Reached Prior to the Massive Leak.**

We wish to highlight that the discussion phase for this paper fully concluded on **Nov 26, 2025**—more than 24 hours before the massive leak incident on Nov 27. The consensus at the close of discussion was **strongly positive (Scores: 10, 6, 6, 6; Avg: 7.0)**.

---

## **Reviewer Consensus (As of Nov 26)**

The reviewers were unanimous in their support, with two reviewers explicitly raising their scores following the rebuttal.


* **5DVv [Score: 6]**: Concerns fully addressed with no further comments.
* **Jnw8 [Score: 6]**: Concerns fully addressed with no further comments.
* **vzx5 [Score: 4 $\to$ 6]**:
> **vzx5 (Nov 23, 2025)**: "Thanks for the authors' **strong response**. I appreciate the key idea of the paper. It's somewhat interesting for me. The authors' response **addressed my core concern** through simplifying the overly complex presentation and the unfounded biological definitions. Besides, the added baseline looks like the **results are good**. I am happy to **raise my score**."
* **ZryQ [Score: 6 $\to$ 10]**
> **ZryQ (Nov 25, 2025)**: "I thank the authors for their **very complete responses** throughout this discussion and I'd like to congratulate them for this **very good work**. I think the paper is **very strong**, built on original and very relevant ideas for the machine learning community, defines clearly its concepts, compares transparently to the state of the art and could **deserve being highlighted at the conference**. I updated my grade."

---

## **Key Revisions & Experiments**

The positive consensus was achieved through a comprehensive rebuttal, incorporating **9 new experiments** and **2 new sections** that explicitly resolved all reviewers' concerns.


|Category|Reviewer Concern (Weakness/Question)|New Section / Experiment|Revision Details|
|-|-|-|-|
|Performance|`vzx5` (W1, W3), `ZryQ` (W1, Follow-up 1, 3)|Section 5.2, Appendix C.8|Table 1, 4, 5, and 6 include a **new SOTA baseline PhyloLM**, showing LLM DNA's consistent and significant superiority even in "same-tokenizer" case.|
|Robustness|Robustness to sentence encoders—`5DVv` (W1, Q3), `Jnw8` (Q3)|Section 5.2, Appendix C.3|Updated **Table 1** demonstrates DNA's robust performance across different sentence encoders. New **Figure 8** shows different sentence encoders lead to correlated DNAs.|
||Robustness to embedding output dimension $p$—`5DVv` (W1, Q3)|Appendix C.4|New **Figure 9** shows DNA quality is marginally affected by embedding output dimension $p$.|
||Robustness to DNA dimension $L$—`5DVv` (Q4), `Jnw8` (Q1)|Appendix C.5|New **Figure 10** shows DNA quality increases as $L$ increases and converges at $L=128$, justifying our hyperparameter choice.|
||Robustness to dataset selection—`5DVv` (W1), `Jnw8` (W1), `ZryQ` (Q1)|Appendix C.7|New **Table 5** validates that DNA extraction works effectively even using random input data, proving resistance to benchmark contamination.|
|Sensitivity|Impact of fine-tuning on DNA—`Jnw8` (Q2)|Section 5.6|New **Figure 6** visualizes the effect of fine-tuning; shows distance generally increases monotonically with data size while global structure remains stable.|
||Impact of chat template on DNA—`ZryQ` (Follow-up 2)|Appendix C.9|New **Table 7** demonstrates that removing chat templates improves the quality of DNA.|
|Fairness|Quality bias on LLM size—`Jnw8` (Q3)|Appendix C.6| New **Table 4** demonstrates DNA quality is statistically independent ($p$-value$>0.6$) of the LLM's parameter size.|
|Discussion|Significance of LLM DNA—`5DVv` (W3, Q1)|Appendix D|New **Appendix D (Broader Impact)** details the utility of DNA for License Governance, Model Selection, and Risk Mitigation.|
||Limitation of LLM DNA—`ZryQ` (W3)|Appendix E|New **Appendix E (Limitations)** discusses traits in DNA subsequences, adaptive attacks, and potential data bias.|
|Presentation|Finite-set assumption—`5DVv` (W4)|Section 3.1|Updated **Section 3.1** clarifies that the finite inputs are a physical fact.|
||Excessive math formulation and biological framing—`vzx5` (W1)|Section 3|Updated **Section 3** simplifies the math formulation and reframed biological analogies into ML terms.|
||More related work—`vzx5` (W2), `ZryQ` (W1)|Section 2|**Related Work** is expanded to better contextualize PhyloLM and representation learning.|
||Definition of evolution—`ZryQ` (W2)|Appendix A.1|New **Definiton A.1** formally defines *Evolution* in the LLM context.|

---

## **Final Remark**
Reviewer feedback significantly strengthened the paper, notably through the **PhyloLM baseline** (demonstrating superiority) and **random-data ablation** (confirming robustness). The discussion concluded **Nov 26** with a **unanimous positive consensus (Avg: 7.0)**, anchored by a **Strong Accept (10)** from Reviewer `ZryQ`, who stated the work **"could deserve being highlighted"**. We remain available to address any further questions or concerns from AC.

---

### Meta-Review · Area_Chair_REJj · 2026-01-07

**Summary:**

This paper proposes LLM DNA, a training-free, low-dimensional functional representation of large language models that enables principled comparison, lineage analysis, and scalable model management. Across reviewers, there is strong agreement that the core idea is original, timely, and technically sound, with particular strength in the bi-Lipschitz formulation and the ability to compare heterogeneous, black-box LLMs without retraining.

Reviewer 5DVv’s key concern was whether DNA quality depends heavily on the choice of sentence embedding backbone; the rebuttal added robustness experiments across multiple encoders and dimensions, which substantially strengthened the empirical case. Reviewer Jnw8 mainly questioned whether DNA reflects only the prompt set and might become “domain-specific DNA”; the authors addressed this with additional dataset-robustness evidence (including random-input extraction) and clarified the intended interpretation as a relative relational signature. Reviewer vzx5’s main critique was that the presentation over-framed generic random projection with heavy math/biology language; the revision simplified the exposition, reduced the biological framing, and improved contextualization and baselines. Reviewer ZryQ’s representative issue was the fairness/clarity of the comparison to PhyloLM; the authors added targeted experiments and revised phrasing to transparently acknowledge comparison limitations, which resolved the concern.

Overall, the revision directly addressed the major points raised, improved transparency and baselines, and leaves the work in a position for acceptance.

**Reviewer Concerns:**

- 5DVv

Concerns about robustness to sentence encoders, embedding dimension, dataset bias, and unclear practical value were fully addressed through new ablations, robustness experiments, and expanded discussion of real-world use cases. No major concerns remain.

- Jnw8

Questions about dataset sensitivity, convergent evolution, fine-tuning effects, and model size dependence were addressed with new experiments showing robustness to random data, predictable DNA shifts under fine-tuning, and statistical independence from model size.

- vzx5

Concerns about over-framing and weak baselines were addressed by simplifying the presentation and adding strong PhyloLM comparisons; the reviewer explicitly stated their core concerns were resolved.

- ZryQ

Initial concerns about missing related work, unclear evolution definition, baseline fairness, and missing limitations were all addressed with new experiments, formal definitions, clearer comparisons, and added limitations; the reviewer raised their score to a strong accept.

**Reviewer Scores:**

The authors left a note on the score changes that reviewers made before the identity leaks in OpenReview.  Given the strong rebuttals, AC believes the following score changes could be realistic.

5DVv: 6 → 6 (concerns resolved, remained positive)

Jnw8: 6 → 6 (concerns resolved, remained positive)

vzx5: 4 → 6 (explicitly raised after rebuttal)

ZryQ: 6 → 10 (explicit strong accept after rebuttal)

---

### Decision · Program_Chairs · 2026-01-26

Accept (Oral)